# Diagnosing ozone-NO$_x$-VOC sensitivity and revealing causes of ozone increases in China based on 2013-2021 satellite retrievals

Jie Ren[1], Fangfang Guo[1], and Shaodong Xie[1]

[1]State Key Joint Laboratory of Environmental Simulation and Pollution Control, College of Environmental Sciences and Engineering, Peking University, Beijing, 100871, China

*Correspondence to*: Shaodong Xie (sdxie@pku.edu.cn)

**Abstract.** Particulate matter (PM$_{2.5}$) concentrations in China have decreased significantly in recent years, but surface ozone (O$_3$) concentrations showed upward trends at more than 71% of air quality monitoring stations from 2015 to 2021. To reveal the causes of O$_3$ increases, O$_3$ production sensitivity is accurately diagnosed by deriving regional threshold values of satellite tropospheric formaldehyde-to-NO$_2$ ratio (HCHO/NO$_2$), and O$_3$ responses to precursors changes are evaluated by tracking volatile organic compounds (VOCs) and NO$_x$ with satellite HCHO and NO$_2$. Results showed that the HCHO/NO$_2$ ranges of transition from VOC-limited to NO$_x$-limited regimes vary apparently among Chinese regions. VOC-limited regimes are widespread found over megacity clusters (North China Plain, Yangtze River Delta, and Pearl River Delta) and concentrated in developed cities (such as Chengdu, Chongqing, Xi'an, and Wuhan). NO$_x$-limited regimes dominate most of the remaining areas. From 2013 to 2021, satellite NO$_2$ and HCHO columns showed an annual decrease of 3.7% and an increase of 0.1%, respectively, indicating an effective reduction in NO$_x$ emissions but a failure reduction of VOC emissions. This finding further shows that O$_3$ increases in major cities occur because the Clean Air Action Plan only reduces NO$_x$ emissions without effective VOC control. Two cases in Beijing and Chengdu also verified that NO$_x$ reduction alone or VOC increase leads to O$_3$ increases. Based on the O$_3$–NO$_x$–VOC relationship by satellite NO$_2$ and HCHO in Beijing, Chengdu, and Guangzhou, the ozone concentration can be substantially reduced if the reduction ratio of VOCs/NO$_x$ is between 2:1 and 4:1.

## 1 Introduction

China has been dedicating to fighting against air pollution in the past years. Since the Chinese government implemented the Air Pollution Prevention and Control Action Plan in 2013, the ambient concentration of particulate matter with aerodynamic diameter of <2.5 μm ($PM_{2.5}$) has decreased significantly (Jiang et al., 2020; Xiao et al., 2021); however, ozone ($O_3$) pollution remains severe or even continues to worsen at least until 2019 (Lu et al., 2020; Zhao et al., 2020). Ozone pollution has restricted the continuous improvement of China's air quality. Thus, it has attracted widespread attention from the government, scientists, and the public.

Tropospheric ozone is produced by photochemical oxidation of volatile organic compounds (VOCs) in the presence of nitrogen oxides ($NO_x$: $NO + NO_2$) (Atkinson, 2000). However, the $O_3$ responses to $NO_x$ and VOC reduction are not linear. The $O_3$ formation throughout much of the troposphere is largely controlled by the availability of $NO_x$ ($NO_x$-limited), but in regions with high $NO_x$ emissions, such as metropolitan areas, $O_3$ formation can be VOC-limited or transform between regimes (Sillman and He, 2002; KLEINMAN, 1994). A VOC-limited regime means that reducing VOC emissions can result in low $O_3$ production; in the presence of high $NO_x$, reduction of $NO_x$ emissions alleviates ozone titration and even leads to ozone increases (Sillman, 1999). Local $O_3$ formation chemistry determines the effect of changes in VOC and $NO_x$ emissions on ozone concentrations.

Various ground-based observation (Nelson et al., 2021; Tan et al., 2018) and model-based methods (Shen et al., 2021; Wang et al., 2019; Wang et al., 2021) have been applied to study $O_3$-$NO_x$-VOC sensitivity. The field observation methods can provide accurate in-situ diagnoses of $O_3$ sensitivity but are limited in temporal and spatial extent (Wang et al., 2017). Air quality model-based methods provide descriptions across time and space. However, uncertainties mainly from emission inventories, as well as meteorological factors and photochemical reactions reduce the accuracy of simulation results (Liu and Shi, 2021). Based on the continuous global observations of formaldehyde (HCHO) and nitrogen dioxide ($NO_2$) by satellite spectrometers, the space-based HCHO-to-$NO_2$ ratio can be used as an indicator of $O_3$ sensitivity because HCHO/$NO_2$ theoretically reflects the relative availability of $NO_x$ and total organic reactivity to hydroxyl radicals (SILLMAN, 1995). This satellite-based method can provide global-scale photochemical information with good time continuity and little human disturbance, which governments and policymakers are keen to know.

In China, there have been many studies exploring $O_3$-$NO_x$-VOCs sensitivity based on satellite-retrieved HCHO/$NO_2$ (Jin and Holloway, 2015; Li et al., 2021). They are based on thresholds derived from earlier work that used air quality models to link HCHO/$NO_2$ with surface $O_3$ sensitivity over the United States (Martin et al., 2004; Duncan et al., 2010; Choi et al., 2012). However, the model approach can be biased because modeled and observed HCHO columns, $NO_2$ columns, and surface $O_3$ often disagree. Jin et al. (2020) derived the threshold values from observations and found the transition in $O_3$ formation regimes occurs at a higher HCHO/$NO_2$ value than previously determined from models. Also, the range of HCHO/$NO_2$ marking transitional regimes varies regionally because of the differences in meteorological conditions, ozone levels, and precursor composition (Schroeder et al., 2017). The threshold extracted in the United States may not be suitable

for China and a uniform value may not be applicable to all Chinese cities. In addition, most previous studies utilized the OMI satellite $HCHO/NO_2$ (Li et al., 2021; Wang et al., 2021). The newly launched TROPOMI aboard Sentinel-5P provides a new perspective to characterize the chemistry of surface $O_3$ at finer spatial scales (Veefkind et al., 2012). Therefore, for a more accurate diagnosis of the ozone-precursor sensitivity across China, the transitional regime range of $HCHO/NO_2$ using the new generation of satellites requires further in-depth investigation.

$O_3$–$NO_x$–VOC sensitivity and changes in VOC and $NO_x$ emissions directly affect ozone concentrations. Nationwide annual estimates of anthropogenic VOC and $NO_x$ emissions in China show different trends from 2013 to 2019, with successive large decreases in $NO_x$ but only slight changes in VOC (Zheng et al., 2018a; Simayi et al., 2022). Previous studies showed that the rapid reduction of $NO_x$ emission increases the surface ozone levels in urban regions (Ren et al., 2021; Yang et al., 2019). In addition, some model simulations suggested that the notable drop of $PM_{2.5}$ is the most crucial factor leading to the $O_3$ increment in China because of the release of $HO_2$ from its uptake on $PM_{2.5}$ (Li et al., 2019). However, this idea remains controversial (Tan et al., 2020). In order to reveal the relationship between ozone increases and changes in its precursors' emissions, satellite-based $NO_2$ and HCHO can be used as visual, timely, and high-resolution trackers of $NO_x$ and VOCs emissions, respectively (Duncan et al., 2010).

In this study, we first take advantage of the Chinese extensive air quality monitoring network to assess the latest trends in ozone concentrations comprehensively. The threshold values that mark the $O_3$ formation transition are derived and used to identify the $O_3$–$NO_x$–VOC sensitivity by matching satellite-based $HCHO/NO_2$ with ground-based $O_3$ measurements across China and in different regions. The effects of precursors on ozone and the causes of ozone increase are explored by combining ozone production regimes with interannual variations in $NO_x$ and VOC emissions inferred from tropospheric $NO_2$ and HCHO vertical columns. Special measures for plague prevention and work resumption during the COVID-19 pandemic also provide an opportunity to assess $O_3$ response to changes in VOC and $NO_x$ emissions. Furthermore, the optimal VOCs/$NO_x$ reduction ratio is probed in three typical cities, Beijing, Chengdu, and Guangzhou, based on the relationships between $O_3$ and $HCHO/NO_2$, to provide scientific support for future $O_3$ pollution control.

## 2 Data and Methodology

### 2.1 Ground-Based Observation of $O_3$ and $NO_2$

The Chinese nationwide air quality monitoring network provides surface $O_3$ and $NO_2$ data. Since 2015, more than 1400 sites have covered more than 330 cities across the country. The nationwide hourly $O_3$ and $NO_2$ concentration data in Chinese cities in 2014–2021 were obtained from the National Urban Air Quality Real-Time Publishing Platform (https://air.cnemc.cn:18007/).

$O_3$ and $NO_2$ measurements were reported in $\mu g\ m^{-3}$ and the mass concentrations for all years were unified to the reference state (298.15 K, 1013.25 hPa). Extensive data quality controls were applied to eliminate unreliable observed outliers following the method of previous studies (Lu et al., 2018). The average of daily $NO_2$ and the daily maximum eight-

hour average (MDA8) $O_3$ concentration for all national-controlled sites in a city was regarded as the city's daily $NO_2$ and $O_3$ levels.

## 2.2 Satellite Observations of $O_3$ Precursors

Given the short lifetime of $NO_x$, the high $NO_2/NO_x$ ratio in the boundary layer, and the fact that HCHO is an intermediate in the oxidation reactions of various VOCs and is approximately proportional to the total rate of VOC reactions with OH radicals (Duncan et al., 2010), current satellite-based spectrometers have provided continuous global observations for two species indicative of $O_3$ precursors, namely, $NO_2$ for $NO_X$ (Lamsal et al., 2014; Martin, 2003) and HCHO for VOCs (Shen et al., 2019; Zhu et al., 2014).

Satellite products of tropospheric $NO_2$ and HCHO vertical columns are retrieved from two satellite instruments: Ozone Monitoring Instrument (OMI) and Tropospheric Monitoring Instrument (TROPOMI). The high spatial resolution ($24 \times 13$ $km^2$ for OMI and $5 \times 3.5$ $km^2$ for TROPOMI) allows for the observation of fine details of atmospheric parameters. They provide daily global observations, and the overpass time (13:40–13:50 and 13:30 local time) is well suited to detect the $O_3$ formation sensitivity when $O_3$ photochemical production peaks, the boundary layer is high, and the solar zenith angle is

small, thereby maximizing instrument sensitivity to HCHO and $NO_2$ in the lower troposphere (Jin et al., 2017).

The OMI data used for 2013-2020 are developed under the Quality Assurance for Essential Climate Variables (QA4ECV) project. The vertical profiles used for QA4ECV products are obtained from the TM5-MP chemical transport model (Williams et al., 2017). The spatial resolutions of OMI $NO_2$ and OMI HCHO are 0.125° and 0.05°, respectively. The TROPOMI data from the Earth Engine Data Catalog are based on the algorithm developments for the QA4ECV reprocessed

dataset for OMI and have been further optimized. The TROPOMI data, available for 2019-2021, are processed with a spatial resolution of 0.01°. The same chemistry transport model for HCHO and $NO_2$ is better suited for analyzing their ratio than products developed with different prior profiles.

OMI data with longer time horizons (2013-2020) are used to study the long-term changes in $NO_2$ and HCHO through their monthly averages and track changes in emissions of $NO_x$ and VOCs. Since the OMI data retrieved by the QA4ECV

project are available until December 2020, the data for 2021 are obtained by converting the TROPOMI data, which is based on a comparison of OMI and TROPOMI monthly data for 2019-2020 to avoid differences in any instrumental offset.

## 2.3 Connecting Satellite HCHO/$NO_2$ with Ground-Based $O_3$ Observations

The high spatial resolution TROPOMI data is used to derive the HCHO/$NO_2$ threshold values marking transitions in $O_3$ formation regimes and to diagnose the current ozone-$NO_x$-VOC sensitivity in China. TROPOMI data are aggregated by

sampling gridded $NO_2$ and HCHO columns consistent with the ground-based observations of $O_3$. $O_3$ concentrations are averaged at all available sites every half month for the warm season (April to September) from 2019 to 2021. $O_3$ measurements at 13:00 to 14:00 local time are selected to match the TROPOMI overpass time. Considering the data volume and the need to reduce retrieval noise, TROPOMI-retrieved $NO_2$ and HCHO columns are sampled every half month

over $O_3$ sites for the same period, yielding nearly 100 000 paired observations. In addition, considering the amount of available data and topographic conditions, China is divided into nine regions to evaluate the satellite-based HCHO/NO$_2$ and study the $O_3$ production regimes, as shown in Fig. S2. Based on satellite and surface observations, the response of $O_3$ concentrations and changes in $O_3$ precursors are compared in two cities, Beijing and Chengdu, during the plague prevention of COVID-19 pandemic and the work resumption in spring 2020 to assess the effectiveness of the $O_3$ production regimes for satellite HCHO/NO$_2$ capture.

# 3 Results and Discussion

## 3.1 Surface Ozone Trends

The ozone concentrations averaged from 367 cities with monitoring sites in China from 2015 to 2021 are shown in Fig. 1a. The three metrics include the maximum 6-month MDA8 ozone running average (MMA6-MDA8) and the 90th and 99th percentiles of the annual distribution of MDA8-O$_3$ (MDA8-90th, MDA8-99th), following the definition in World Health Organization global air quality guidelines (World Health Organization, 2021) and Chinese ambient air quality standards (GB 3095-2012). The three metrics increased at 2.3 (2%), 2.5 (2%), and 2.3 (1%) µg m$^{-3}$ year$^{-1}$ in 2015-2021. Moreover, they significantly increased at 4.6 (5%), 5.6 (4%), and 5.4 (3%) µg m$^{-3}$ year$^{-1}$ (all with $p < 0.05$) in 2015-2019, based on the linear trend estimate.

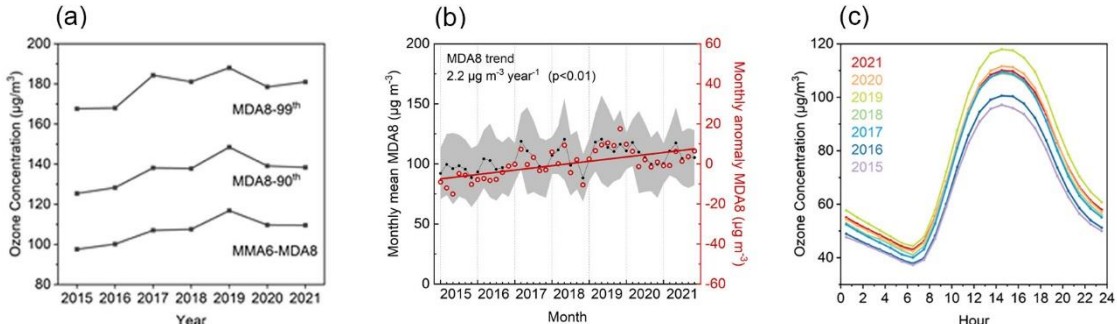

Figure 1: Surface ozone concentrations in China. (a) MDA8-99th, MDA8-90th, MMA6-MDA8, and (b) monthly average (black dots, left axis) and anomaly (circle, right axis) of MDA8 ozone concentrations averaged from all 367 cities with monitoring sites in 2015–2021. Gray shading: mean value ± standard deviation across all cities for each month. Solid line: linear fitted curve. (c) Ozone diurnal cycles from April to September.

The time series of the monthly mean MDA8 ozone in 2015–2021 is shown in Fig. 1b. Based on the monthly anomalies (the difference between the observed monthly mean and the 7-year monthly mean for the same month), MDA8 ozone levels increased by 2.3 µg m$^{-3}$ year$^{-1}$ (3.0% year$^{-1}$, $p < 0.01$) averaged from all cities. The increases are also exhibited rapidly before 2019, with 3.5 µg m$^{-3}$ year$^{-1}$ (4.5% year$^{-1}$, $p < 0.01$). The daytime and nighttime mean ozone levels show similar trends. In 2015–2019, the maximum and minimum hourly average concentration increased from 97.1 to 117.9 µg m$^{-3}$ and

from 37.3 to 44.3 μg m⁻³, respectively, based on the mean April to September ozone diurnal variation, as shown in Fig. 1c.

Overall, the average ozone concentration shows an increasing trend from 2015 to 2021, although it has decreased in the last two years compared to the peak in 2019.

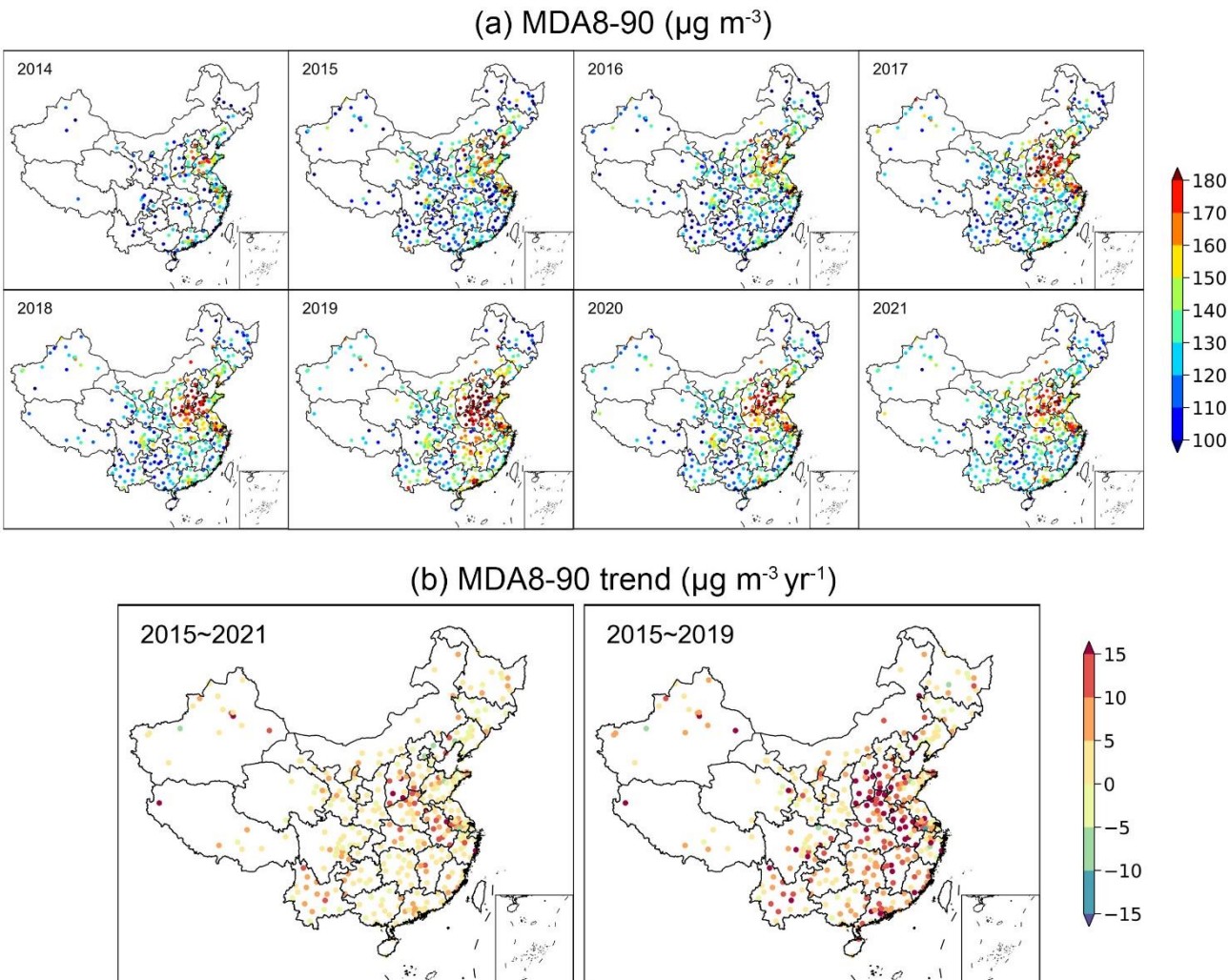

**Figure 2: Spatiotemporal distributions of MDA8-90th ozone in China from 2014 to 2021, and trends for 2015–2019 and 2015–2021.**

The spatial patterns of MDA8-90th and their trends for approximately 367 cities in 2014–2021 are shown in Fig. 2. The cities with high ozone levels are widely distributed in eastern and central China, where intensive anthropogenic emissions are located. Distinct exacerbation of ozone pollution can be observed from 2014 to 2019. The locations and definitions of the key regions are shown in Fig. S1. In 2014–2015, sites with MDA8-90th greater than 160 μg m⁻³ were concentrated near the Beijing-Tianjin-Hebei and the Yangtze River Delta (YRD) megacity clusters. By 2019, ozone hot spots extended westward

to the Fenwei Plain and southward to the central Yangtze River Plain. During the 6 years, the ozone concentration has
increased almost everywhere in China, particularly in the North China Plain (NCP) and the central Yangtze River Plain. In
2020 and 2021, ozone pollution was alleviated; the level of which is equivalent to the level of ozone pollution in 2017–2018.
$O_3$ concentration decreases in major key regions, including the North China Plain, Pearl River Delta (PRD), and Sichuan
Basin (SCB).

The percentage of sites with MDA8-90th ozone higher than 160 μg m$^{-3}$, the Chinese grade II national air quality
standard, increased from 15.0% in 2015 to 39.3% in 2019 and then decreased to 23.0% in 2021. The percentage of sites with
increasing ozone trends was 83.1% in 2015–2019 and 71.8% in 2015–2021. From a national perspective, widespread surface
ozone increases occurred in Chinese cities during 2015–2019. Moreover, decreases have been observed in the last 2 years.

### 3.2. Ozone–NO$_x$–VOC sensitivity over China

### 3.2.1 Ozone–NO$_x$–VOC Chemistry Captured by Satellite-Based HCHO/NO$_2$

Ozone formation can be limited by NO$_x$, VOCs, or both. Satellite-based HCHO and NO$_2$ columns are evaluated
whether they can capture the nonlinearities in O$_3$–VOCs–NO$_x$ chemistry. Figure 3a, similar to classic ozone isopleths
typically generated with models (Sillman et al., 1990; Pusede et al., 2015), shows the in situ O$_3$ concentration as a function
of TROPOMI NO$_2$ and HCHO derived solely from observations.

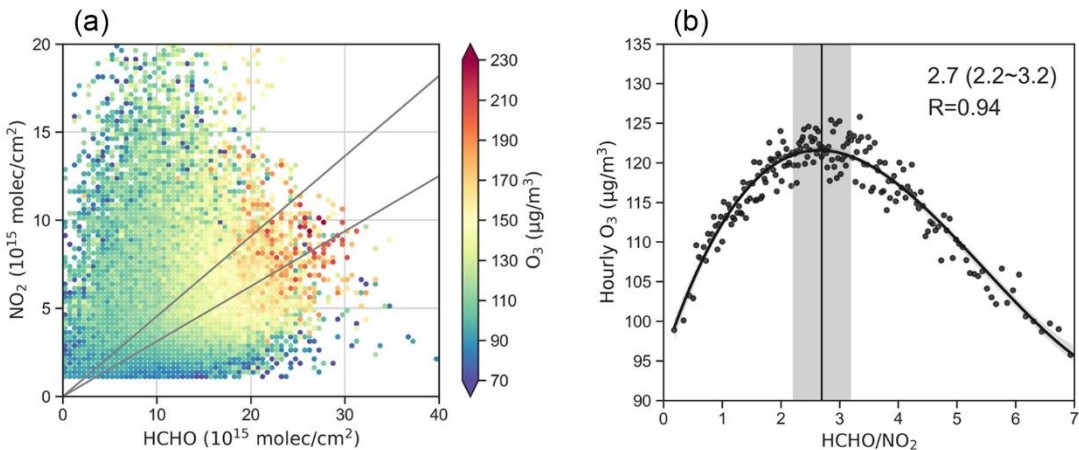

**Figure 3: (a) O$_3$ concentration as a function of TROPOMI HCHO and NO$_2$. All surface hourly O$_3$ observations (13:00 to 14:00
local time) averaged over each half month from April to September 2019–2021 are aggregated based on the corresponding
TROPOMI HCHO and NO$_2$ (interval: 0.5 unit × 0.25 unit). The black lines delineate the TROPOMI HCHO/NO$_2$ values of 2.2 and
3.2. (b) O$_3$ concentration (13:00 to 14:00) as a function of TROPOMI HCHO/NO$_2$. O$_3$ average concentration is calculated by first
matching hourly O$_3$ observations with TROPOMI HCHO/NO$_2$ for each half of the month, dividing these paired observations into
200 bins based on HCHO/NO$_2$, and then calculating the O$_3$ average concentration (_y_ axis) for each TROPOMI HCHO/NO$_2$ bin
(_x_ axis). The solid line is fitted with third-order polynomial curves, and the shading indicates 95% confidence intervals. The
vertical lines indicate the maximum of the fitted curve, and the vertical shading represents the range with the curve slope from −3
to +3 (regime transition).**

Consistent with $O_3$ isopleths, $O_3$ concentration is a non-linear process in relation to $NO_2$ and HCHO, as shown in Fig. 3a. Three regimes can be roughly identified. When HCHO is low and $NO_2$ is relatively high, the $O_3$ concentration is high at lower $NO_2$ or higher HCHO, indicating VOC-limited (or $NO_x$-saturated) chemistry. When $NO_2$ is low and HCHO is relatively high, the $O_3$ concentration increases with increasing $NO_2$, indicating $NO_x$-limited chemistry. When both $NO_2$ and HCHO are high, $O_3$ concentration peaks and increases with both increasing $NO_2$ and HCHO. However, the division between the three regimes is uncertain and ambiguous, which may be influenced by other factors such as different meteorology, noisy satellite retrievals, the spatial mismatch between gridded satellite observations and point measurements on the ground, and lack of statistical power to calculate average concentrations in some intervals due to small sample sizes. Despite these uncertainties, Figure 3a qualitatively illustrates the nonlinear relationship between $O_3$ and satellite $NO_2$ and HCHO, similar to the overall $O_3$-$NO_x$-VOC chemistry.

After the qualitative approach is established, the HCHO/$NO_2$ thresholds that mark the $O_3$ transitional regime are then derived by calculating the average ozone concentration for a given TROPOMI HCHO/$NO_2$ from Fig. 3a and examining their statistical relationships in China and the different regions within it. Empirical relationships are investigated by applying second-order and third-order polynomial models (Jin et al., 2020) to the observations over all cities in China. The third-order polynomial model is used to derive the maximum mean $O_3$ concentration (the peak of the curve in Fig. 3b) because it fits the data with the high correlation coefficient better than the second-order model does. Assuming that the peak of the curve (with a slope of 0) marks the transition from VOC-limited to $NO_x$-limited regimes, the transitional regime (mixed sensitive regime) is defined as the range with a slope between −3 and +3.

The aggregation of the overall available observations used in Fig. 3a shows that the $O_3$ concentration peaks at HCHO/$NO_2$ are equal to 2.7, with the transitional regime between 2.2 and 3.2. A separate evaluation for nine regions also reveals the strong nonlinear relationships between the $O_3$ concentration and TROPOMI HCHO/$NO_2$, despite differences in the overall $O_3$ concentrations. The Pearson correlation coefficient between the fitted third-order polynomial curve and the data is generally more than 0.73, except for Inner Mongolia (0.59). The HCHO/$NO_2$ marking the regime transition varies among these regions (Table 1 and Fig. S3), with the highest in Shanxi-Shaanxi-Henan [3.7 (3.2–1.3)] and the lowest in Heilongjiang-Jilin-Liaoning [2.0 (1.2–3.2)]. Different cities respond differently to changes in $NO_x$ and VOC emissions. Previous studies have also demonstrated regional differences in the threshold values of HCHO/$NO_2$ (Schroeder et al., 2017; Chang et al., 2016) and other photochemical indicators (Liu and Shi, 2021). This difference among regions may reflect environmental conditions (Liu and Shi, 2021), such as radiation intensity and surface temperature.

**Table 1. HCHO/NO$_2$ threshold values between O$_3$ production regimes in nine Chinese regions**

| Region | Correlation coefficient [a] | HCHO/NO$_2$ [b] | Transitional regime |
|---|---|---|---|
| Beijing-Tianjin-Hebei-Shandong | 0.93 | 3.4 | 3.0–3.8 |
| Shanxi-Shaanxi-Henan | 0.93 | 3.7 | 3.2–4.3 |
| Shanghai- Jiangsu-Zhejiang-Anhui | 0.85 | 2.7 | 2.2–3.3 |
| Jiangxi-Hubei-Hunan-Fujian | 0.76 | 3.2 | 2.2–4.7 |
| Guangdong-Hong Kong-Macao- Guangxi-Hainan | 0.82 | 3.2 | 2.7–3.8 |
| Sichuan-Chongqing- Guizhou-Yunnan | 0.73 | 3.2 | 2.5–4.3 |
| Gansu-Ningxia-Qinghai-Tibet-Xinjiang | 0.74 | 3.0 | 2.2–4.5 |
| Inner Mongolia | 0.59 | 2.5 | 1.9–3.4 |
| Heilongjiang-Jilin-Liaoning | 0.82 | 2.0 | 1.2–3.2 |

[a] The Pearson correlation coefficient between the fitted third-order polynomial curve and the data.

[b] HCHO/NO$_2$ value at the maximum of the fitted curve (peak O$_3$ concentration).

The HCHO/NO$_2$ thresholds in the present study are higher than the previously reported model-based values, such as 1~2 by Jin and Holloway (2015) and Duncan et al. (2010), and 1.5~2.3 by Chang et al. (2016). Thus, this observation-based method indicates that O$_3$ production tends to be VOC limited at the same HCHO/NO$_2$. This discrepancy stems from the different approaches linking HCHO/NO$_2$ to O$_3$ production regimes. Previous modelling studies derived thresholds by simulating the response of surface O$_3$ to the overall reductions in NO$_x$ or VOC emissions. Moreover, these thresholds were derived from the study area in the United States, which best represents regional rather than local O$_3$-NO$_x$-VOC sensitivities (Martin et al., 2004; Jin et al., 2017). Thresholds derived with in situ observations in China take full account of the local O$_3$ chemistry.

### 3.2.2 Spatial Distribution of O$_3$ Production Regimes

The derived HCHO/NO$_2$ thresholds were used to identify the O$_3$ production regimes over China from April to September 2019–2021. Based on the same HCHO/NO$_2$ threshold across China (2.2–3.2) and different thresholds for different regions, two methods yield similar spatial distributions of O$_3$ sensitivity (Fig. S4 and 4). However, the regional approach suggests that O$_3$ production tends to be VOC limited because the HCHO/NO$_2$ thresholds derived in many key regions individually are higher than the national overall threshold.

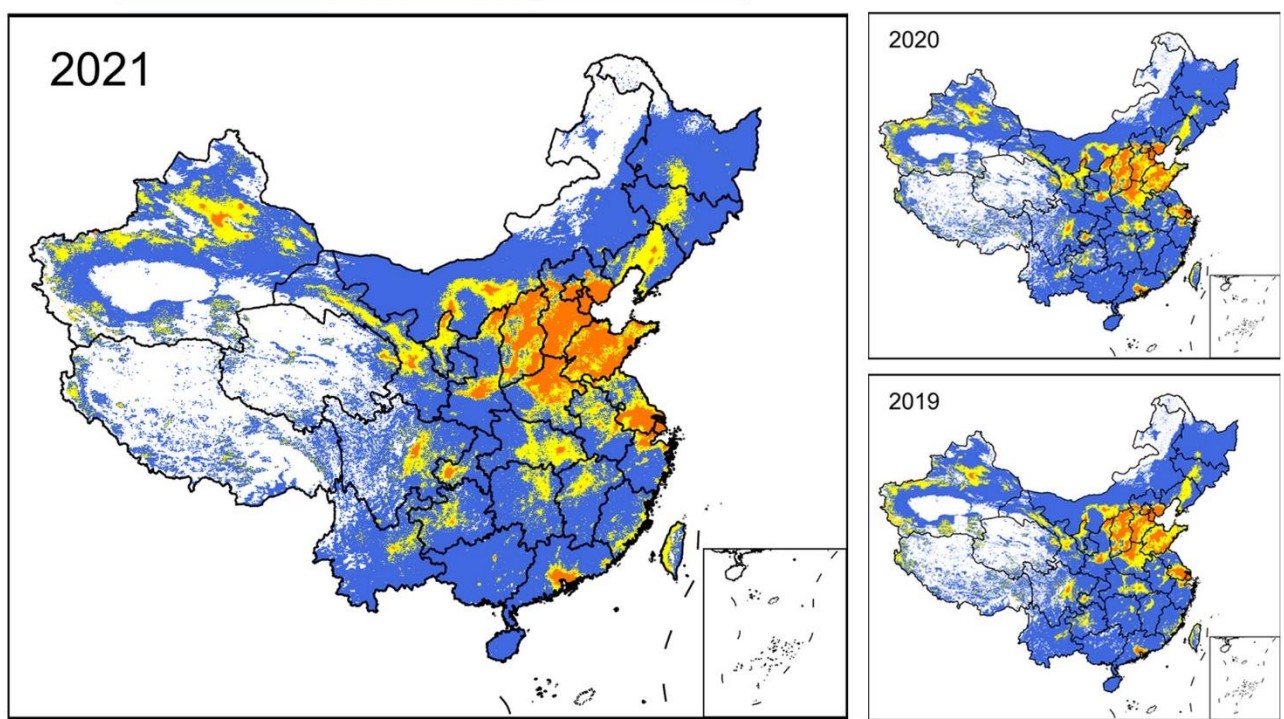

**Figure 4: Ozone sensitivity classification over China from April to September 2019–2021 using different HCHO/NO₂ thresholds in different regions. Only polluted regions are displayed (defined as average TROPOMI NO₂ columns higher than $1.0 \times 10^{15}$ molecules/cm$^2$).**

As shown in Fig. 4, the NO$_x$-limited regime dominates in most regions from April to September. VOC-limited chemistry exists in urban areas of almost every province, with transitional regimes occurring around the VOC-limited regimes in wide urban and suburban areas. Widespread VOC-limited and transitional regimes are observed over major megacity clusters. In the North China Plain, the VOC-limited regimes are distributed in Beijing and most regions in the provinces of Hebei, Shanxi, Henan, and Shandong. Transitional regimes control almost all the remaining regions, except for individual cities in northern Hebei and southwestern Henan. The percentages of VOC sensitive regime and the transitional regime are 59.2% and 26.7% in 2021 (Table S1). In the Yangtze River Delta, the VOC-limited regimes are found in some developed urban cities including Shanghai, some cities in southern Jiangsu Province (such as Suzhou, Wuxi, and most of Nanjing and Nantong) and Zhejiang Province (Jiaxing, urban of Hangzhou and Ningbo). VOC-limited and transitional or mixed regimes occupy 23.5% and 31.4% of the total grid cells (Table S1). In the Pearl River Delta and surrounding areas (Guangdong Province), VOC-limited regimes (8.0%) and transitional regimes (11.0%) control Shenzhen, Dongguan, Foshan, Zhongshan, Zhuhai, most of Guangzhou, and some other urban areas, whereas NO$_x$-sensitive regimes dominate most of the remaining cities. In addition, extensive VOC-limited and transitional regimes occur in developed cities of Sichuan Basin

(urban centres of Chongqing and Chengdu), Shaanxi Province (Xi'an), Hubei Province (Wuhan), and other regions. Compared with previous studies (Jin and Holloway, 2015; Wang et al., 2019), the results of this study are slightly different. The areas dominated by VOC-limited and transitional regimes in China are expanded, especially in NCP; while in the southern region (e.g., Guangdong Province), VOC-sensitive regimes are only concentrated in major cities.

Air quality monitoring sites are generally located in built-up areas of cities, representing the environment in which people live. The ozone sensitivity regime of all monitoring sites' locations based on HCHO/NO$_2$ and the O$_3$ MDA8-90th in 2021 are shown in Fig. S5. VOC-limited and transitional regimes dominate O$_3$ formation at 86.2% of the sites, with VOC-limited chemistry accounting for 52.5%.

The comparison of O$_3$ sensitivities from 2019 to 2021 (Fig. 4) shows a slight change in the trends of individual regions
from VOC-limited regimes to transitional regimes or from transitional regimes to NO$_x$-limited regimes as a result of NO$_x$ reduction. However, there has been no remarkable change in China as a whole. The changes in VOC and NO$_x$ emissions in recent years have not led to the transition in the spatial distribution of the O$_3$ production regime defined by our HCHO/NO$_2$ thresholds.

### 3.3 Effects of Ozone Precursors Variations on Ozone

### 3.3.1 Interannual Variations in Ozone Precursors

Based on the non-linear O$_3$-NO$_x$-VOC relationship, changes in ozone precursors can directly affect ozone levels. Ground-based measurements of NO$_2$ and satellite-based NO$_2$ columns are used to indicate the changes in NO$_x$ emissions. As shown in Fig. 5a and S6, NO$_2$ is concentrated over urban areas and industrial cities, particularly in megacity clusters, such as the NCP, YRD, and PRD, and large cities, such as Wuhan (in Hubei) and Xi'an (in Shaanxi). Satellite observations show
large NO$_2$ decreases from 2013. This finding is consistent with previous studies (Wang et al., 2019; Lin et al., 2019). Moreover, the decrease is more than 2.8% per year in half of the grids. Figure 5b shows that the average annual reduction in satellite NO$_2$ in eastern China is 3.7% ($0.18 \times 10^{15}$ molecules cm$^{-2}$, $p < 0.01$). A reverse increase has occurred in the last 2 years, mainly in winter. Figure S8 shows similar trends in surface NO$_2$ concentrations and satellite-based NO$_2$ columns averaged in April–September. More than 75% of the cities show negative trends in surface NO$_2$ levels (and 22% with $p <$
0.05), and nearly 73% of the grids show decreases in satellite NO$_2$ columns. Strong and consistent decreases in high-value areas, particularly the NCP, as well as YRD, PRD, and SCB, have been observed in the last 9 years. The mean NO$_2$ in NCP has decreased by $0.59 \times 10^{15}$ molecules cm$^{-2}$ year$^{-1}$ (5.0%, $p < 0.01$) from 2013 to 2021.

The variations in VOCs primarily drive the temporal patterns of satellite HCHO columns over China (Shen et al., 2019). The changes in satellite HCHO can roughly indicate changes in VOC emissions, which has been applied in several previous
studies (Li et al., 2020; Shen et al., 2019; Zhu et al., 2014). Figures 6a and S7 compare the HCHO columns from April to September 2013–2021. As expected, a clear gap exists between the HCHO in southeast and northwest China. The mean HCHO is high over megacity clusters in the southeastern regions, such as the Northern China Plain, Sichuan Basin, Yangtze

River Delta, Pearl River Delta, and the Central Yangtze River Plain. More than 45% of the grids show positive trends in satellite HCHO columns in 2013–2021. HCHO columns increase by 0.01% year$^{-1}$, averaged from all grids in 2013–2021 in eastern China. This finding is consistent with the increases in anthropogenic VOC emissions (Zheng et al., 2018b; Zhang et al., 2019). In 2020–2021, satellite-based HCHO decrease in most of eastern China compared to 2019.

In addition to anthropogenic and biogenic emissions, the long-term changes in HCHO are driven by several other factors (Zhu et al., 2017): NO$_x$ reductions can lower the HCHO yield from isoprene oxidation (Souri et al., 2020; Wolfe et al., 2016), and in addition, HCHO columns may show interannual variability driven by interannual variability of meteorology, particularly temperature as shown in Fig. S9 (Duncan et al., 2009; Abbot et al., 2003). Nevertheless, HCHO has not declined as dramatically as NO$_2$, particularly until 2019. In the NCP, where VOC-limited chemistry most widely exists, the gap between HCHO and NO$_2$ changes is even larger.

(a) Satellite NO$_2$ (molec/cm$^2$)

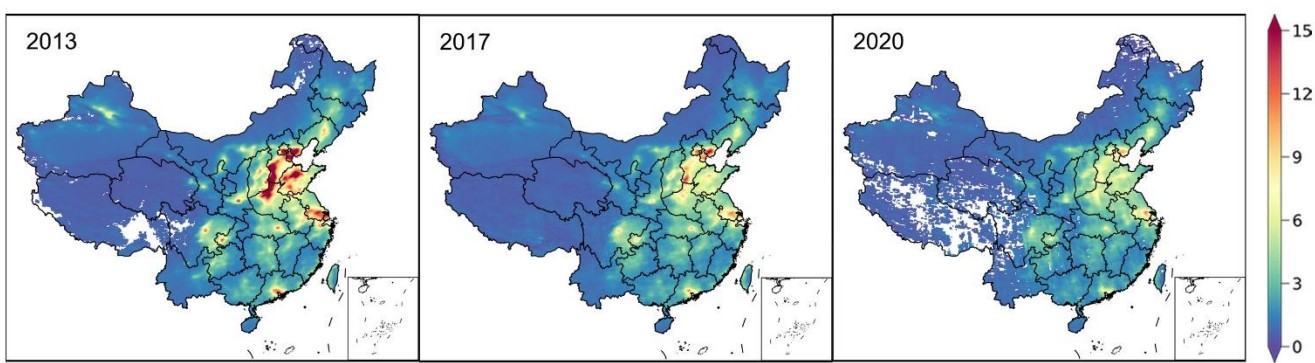

(b)

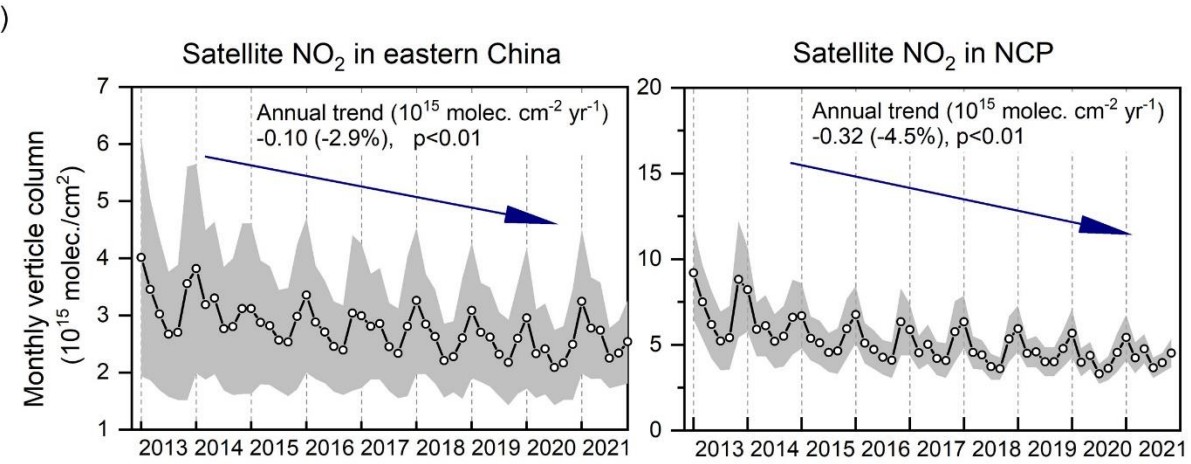

**Figure 5: (a) Maps of average satellite-based NO₂ columns over China from April to September, and (b) monthly mean NO2 columns averaged over eastern China and North China Plain in April-September. Gray shading: mean value ± 50% standard**

deviation across all grids for each month. Solid line: the linear fitted curve. Inset: absolute annual linear trend and percentage of annual trend (% per year, the linear trend divided by the 2013 mean values).

## (a) Satellite HCHO (molec/cm²)

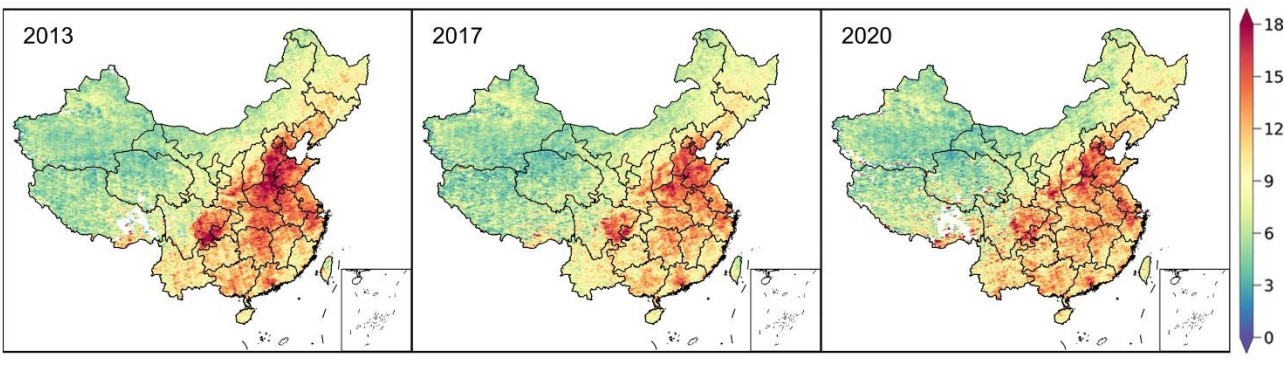

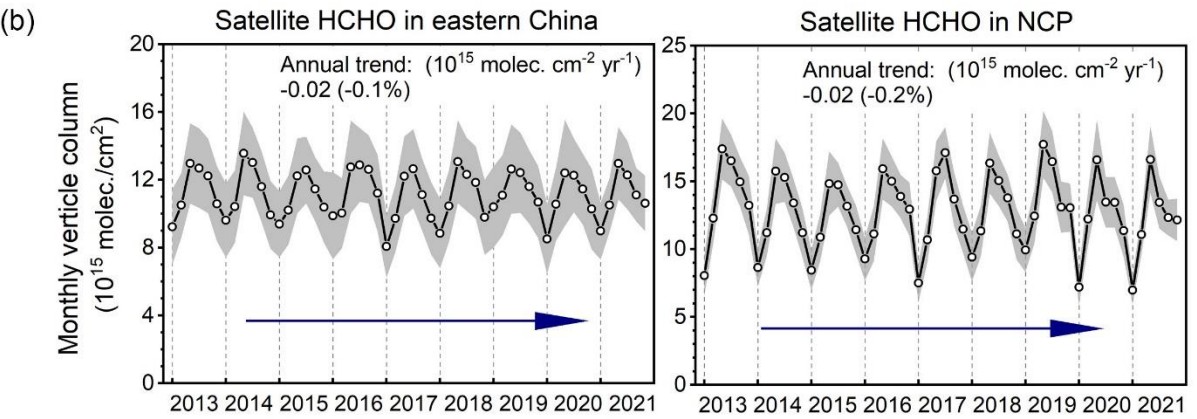

**Figure 6: Same as Figure 5 but for satellite-based HCHO columns.**

The changes in VOC and $NO_x$ emissions presumably have the same disparity as the changes in satellite HCHO and $NO_2$. The anthropogenic $NO_x$ emission has been decreased considerably since 2013. However, VOC emissions have not been effectively reduced or even slightly increased. The significant $NO_x$ reduction alone without effective VOC control has not worked or has even aggravated $O_3$ pollution in recent years because of the VOC-limited (or $NO_x$-saturated) $O_3$ production regimes in major city clusters. Considering that biological VOC also increased (Li et al., 2020), it is clear that the reduction of anthropogenic VOCs is not sufficient to bring down the total VOC emissions.

### 3.3.2 Cases of $O_3$ responses to the changes in its precursors

The outbreak of the COVID-19 pandemic produced previously unseen societal impacts in China. The measures during the plague prevention and the work resumption resulted in changes in VOC and $NO_x$ emissions in 2020 compared to normal

years (Le et al., 2020; Pei et al., 2022), and these changes were not synchronized across Chinese cities. The ozone pollution that occurred in Beijing and Chengdu during this period is used as natural experiments to evaluate surface $O_3$ responses to apparent emission variations in order to validate the conclusions above.

In Beijing, the capital of China, the average $O_3$ MDA8 concentration reached 118.9 μg m$^{-3}$ in April 2020, which is 15.5% and 11.9% higher than that in April 2019 and 2021 (Fig. 7a). The satellite or ground-based observations of $NO_2$ and HCHO show variations in $NO_x$ and VOCs. $NO_2$ concentrations were the lowest in April 2020 compared with the monthly means in April 2019 and 2021, showing reductions of 33.2%–42.1% and 17.8%–28.5%. This scenario is associated with a massive reduction in human activity (Le et al., 2020). Unlike the change in $NO_2$, HCHO columns in April 2020 are only 16.5% lower than those in 2019 and even 8.7% higher than those in 2021. In the $NO_x$-saturated (VOC-limited) regime, VOC emission reductions are generally effective in reducing $O_3$ concentrations. However, the former is primarily offset by the reduction in $NO_x$ emissions resulting in a low titration of $O_3$ by $NO_x$, causing directly the increase in $O_3$.

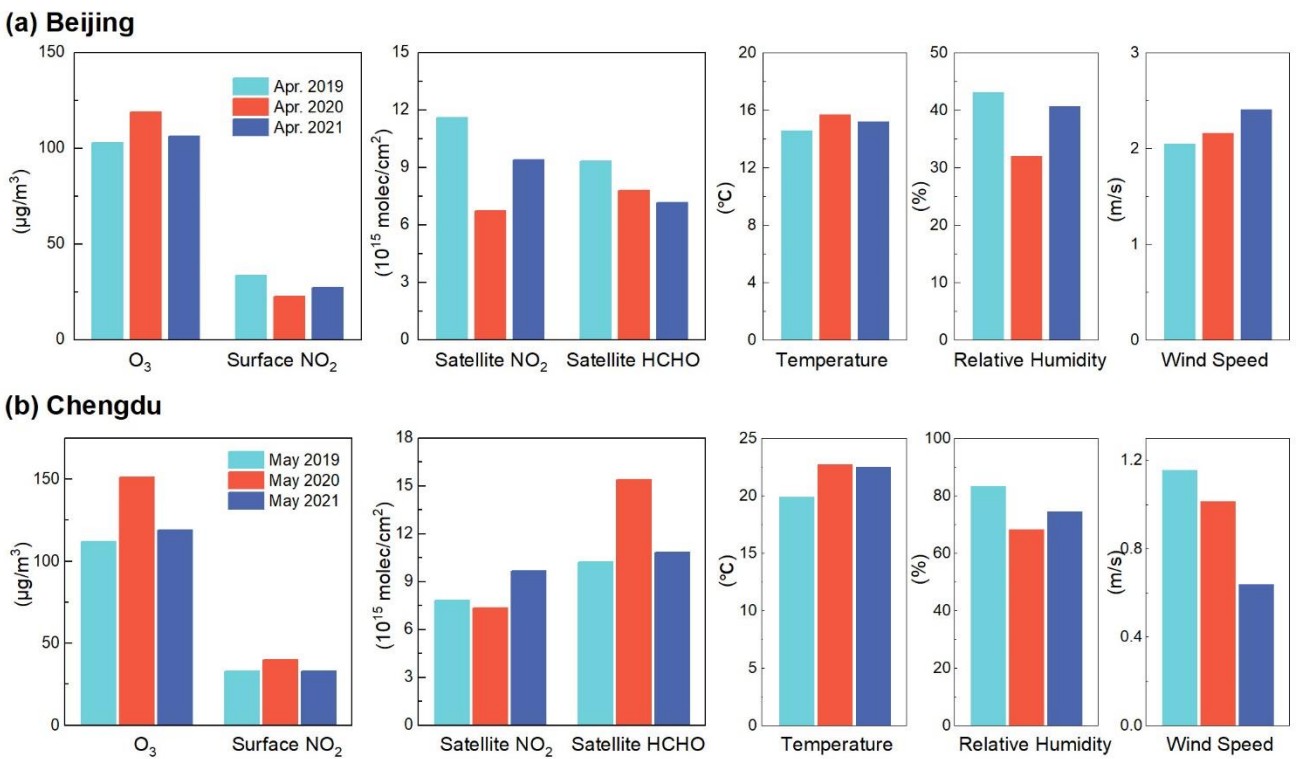

**Figure 7: Ground-based observation of $O_3$ and $NO_2$, satellite-based $NO_2$ and HCHO, and meteorological conditions, including temperature, relative humidity and windspeed in (a) Beijing and (b) Chengdu, Sichuan. The figure compares monthly averages for Beijing in April and for Chengdu in May 2019–2021.**

Severe ozone pollution occurred in Chengdu, the capital of Sichuan Province, in May 2020, with the maximum hourly ozone concentration reaching 258.8 μg m$^{-3}$. The monthly average MDA8 is 34.8% and 26.9% higher than that in May 2019 and 2021, respectively. Compared with the monthly means in May 2019 and 2021, the monthly means in May 2020

demonstrate the almost constant $NO_2$ ($-6.1\%$ to $21.2\%$ and $-24.3\%$ to $21.2\%$) and the considerable increase in HCHO ($50.7\%$ and $42.2\%$), as shown in Fig. 7b, which is associated with the increase in VOC emissions (Song et al., 2022). As expected, the increase in VOC emissions is a major contributor to the growth of $O_3$ because of the VOC-limited conditions in Chengdu.

At that time, after the strictest restrictions, Chengdu was vigorously developing the stall business to help work resumption. The increase in VOCs may be related to open-air cooking emissions (Liang et al., 2022).

In these two cases in Beijing and Chengdu, meteorological conditions also contributed to the increase in ozone concentrations. As shown in Figure 7, compared to the previous and subsequent years, there is only a slight increase in temperature; relative humidity decreases, but remains high in Chengdu (over 60%); wind speed is not the lowest in the three

325 years. Although the effects of emissions and meteorological conditions are difficult to distinguish quantitatively, it reflects the negative effect of $NO_x$ reduction alone or VOC increase on $O_3$ pollution in urban cities.

## 3.4 VOC/NO$_x$ Reduction Ratio for Typical Cities

The synergistic control of NOx and VOC, with VOC control as the focus, can help reduce ozone pollution in major cities. The reduction ratio of VOC/$NO_x$ can be indicated by the slope of the ridge line of the classic ozone isopleths (Ou et al.,

2016; Wang et al., 2019). The relationship between $O_3$ concentrations and the satellite HCHO and $NO_2$ columns reflected by the approach in Figure 3a is similar to the $O_3$-VOC-$NO_x$ isopleths. Thus the peak of the fitted curve of $O_3$ and HCHO/$NO_2$, corresponding to the ridge line of $O_3$ isopleths, is the ratio of HCHO to $NO_2$ reduction for the fastest decrease in $O_3$, i.e., $\triangle HCHO/\triangle NO_2$.

This approach is used to provide preliminary scenarios for synergistic VOC and $NO_x$ reduction in three developed cities,

Beijing in the NCP, Guangzhou in the PRD, and Chengdu in the SCB, which are all dominated by the VOC-limited regime. As shown in Fig. 8, the optimal decrease ratio of HCHO to $NO_2$ is between 2:1 and 4:1, which is 3.4–3.9 for Beijing, 2.2–2.6 for Chengdu, and 2.3–2.8 for Guangzhou. The results obtained by this method roughly reflect the optimal abatement pathways for the two precursors by using the satellite HCHO and $NO_2$ to represent VOC and $NO_x$. The short-term future $O_3$ control becomes much more promising in terms of enhanced VOC control in urban cities dominated by VOC-limited

regimes.

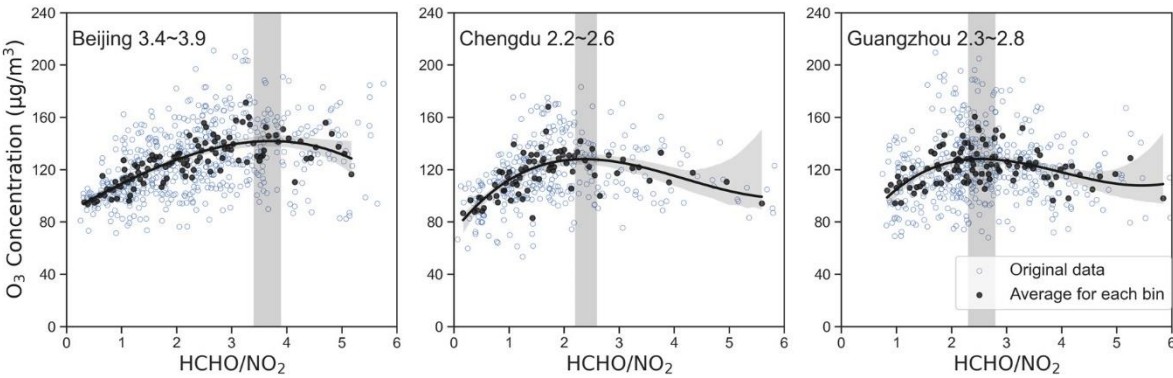

**Figure 8: O₃ concentration as a function of HCHO/NO₂ for all sites in Beijing, Chengdu, and Guangzhou. Black dots and lines are the same as those in Figure 3b but divided into 100 bins, and blue circles are original concentrations.**

## 4 Discussion

Based on the evaluation of the nonlinearity of O₃-NOₓ-VOC chemistry captured by space-based HCHO/NO₂, this study derives thresholds marking the transitions between chemical regimes in different regions of China and diagnoses the current spatial distribution of O₃ production regimes. The higher resolution TROPOMI satellite data enables a better match between the location of ground-based O₃ monitoring stations and the grid of satellite data, thus allowing a more accurate derivation of HCHO/NO₂ thresholds and reflection of spatial variations in ozone-NOₓ-VOC sensitivity.

However, the use of satellite HCHO/NO₂ for quantitative diagnosis of ozone-NOₓ-VOC sensitivity is influenced by the following uncertainties and requires further investigation. First, the thresholds derived from the observation-based approach may be affected by biases in the satellite retrieval algorithm. Satellite instruments measure vertically integrated column densities, and the inhomogeneity of the vertical distribution might impose negative influences (Schroeder et al., 2017). Second, the relationship between satellite-based HCHO and surface VOC reactivity is not fully understood. There are

differences in HCHO yields for different classes of VOCs (Shen et al., 2019). Biogenic VOC emissions also have an influence. Since HCHO is a weaker UV-visible absorber than NO₂, satellite retrieval of HCHO is more error-prone. These factors may limit their usefulness in detecting HCHO from local sources of anthropogenic VOCs.

       Identifying the causes of ozone increases since 2013 is crucial for effectively controlling the ozone pollution in China. The Chinese government has been controlling mainly particulate matter, SO₂, and NOₓ since 1998 (Zhang et al., 2016).

The Air Pollution Prevention and Action Plan, implemented in 2013, aims to reduce the annual average PM₂.₅ concentration by 10% in 2017 compared with 2012 (Chinese State Council, 2013). In 2018, the Chinese government promulgated the Three-Year Action Plan for Winning the Battle of the Blue Sky, which still regards PM₂.₅ control as the primary goal and targets SO₂ and NOₓ emission reductions (Chinese State Council, 2018). These nationwide emission control measures have decreased PM₂.₅ concentrations rapidly (Xiao et al., 2021) and reduced anthropogenic NOₓ emissions. However, the decline

in VOC emissions is not yet evident. VOC-limited chemistry exists in most city clusters in China, particularly in built-up

areas of cities where air quality monitoring sites are located. Our study highlights that the root cause of ozone increase in major regions is the significant reduction of $NO_x$ alone without effective control of VOC.

Some studies suggested that the significant decrease in $PM_{2.5}$ is the most crucial factor contributing to the $O_3$ increment in China (Li et al., 2019), but the idea of the impact of $HO_2$ uptake is controversial (Tan et al., 2020). The summertime MDA8 $O_3$ enhancement due to changes in $PM_{2.5}$ levels in NCP during 2013-2017 estimated by Li et al. (1 ppb year$^{-1}$) is insufficient to explain the observed trend (3.3 ppb year$^{-1}$) (Lu et al., 2020), which indicates that the effect of $PM_{2.5}$ is not the essential cause of worsening ozone pollution. The simultaneous decline in $O_3$ and $PM_{2.5}$ concentrations in Beijing in recent years (Ren et al., 2021) suggests that appropriate VOC and $NO_x$ emission reductions can control both $PM_{2.5}$ and $O_3$ pollution.

Currently, the VOC-limited regime exists in most major urban clusters in China, which are also areas with high ozone levels. The exploration of the causes of ozone rise in China illustrates the current need for VOC control, which is consistent with the findings of previous studies on major cities and city clusters (Wang et al., 2022; Yang et al., 2019). Deep $NO_x$ reductions will eventually reduce $O_3$ concentrations to lower levels. However, if only $NO_x$ is currently reduced without VOC reductions, $O_3$ concentrations will not be effectively reduced until $NO_x$ is reduced to very low levels, leading to a shift in $O_3$ formation mechanisms toward $NO_x$-limited. This would take a long time and be very costly. In contrast, synergistic control of $NO_x$ and VOC, with a focus on VOC control, is currently a direct and effective approach to alleviate ozone pollution in large cities.

**Data availability**

The data used in this study can be accessed by contacting the corresponding author, Shaodong Xie (sdxie@pku.edu.cn).

**Author contribution**

SX and JR initiated the research project. JR performed the data analyses and wrote the manuscript. SX and FG reviewed and revised the paper.

**Competing interests**

The authors declare that they have no conflict of interest.

**Acknowledgments**

This work is supported by the National Key Research and Development Program of China (No. 2018YFC0214001).

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
