# Peer review of "Diagnosing ozone-NOx-VOC sensitivity and revealing causes of ozone increases in China based on 2013-2021 satellite retrievals"

_Atmospheric Chemistry and Physics, 2022_

## Referee Comment (RC2)

**Major comments:**

I am concerned that the authors suggest the need to shift from a focus on NOx reduction to VOC reduction. Once NOx reductions cease to result in increased rates of ozone production due to the non-linear response of ozone to NOx, ozone levels will again start to decrease. Thus continued NOx reductions will eventually reduce ozone even without reducing VOCs. Certainly there are plenty of reasons to reduce VOCs, but continued NOx reductions will eventually reduce ozone and efforts should not move away from reducing NOx. The authors could consider calculating at what point that might start to occur. This paper, Wang et al., 2022 - https://doi.org/10.5194/acp-22-8935-2022 – used box modeling to show that cities transitioned from VOC-limited to a transition regime, and concluded that China needs concurrent reductions in both VOC and NOx reductions. The authors need to address the findings of this paper, and differentiate their analysis, which is similar in assessing the long-term trends in ozone.

The authors do not provide a convincing argument that the increase in satellite HCHO is due to a lack of emissions reductions. According to Zheng et al., 2018a referenced by the authors, the VOC composition has changed significantly with reductions in residential and transportation VOCs but increases in industrial and solvent VOCs. The authors need to discuss the impact of the change in VOC composition, particularly any shift in yield of HCHO from the change in VOC mixture and the reduction in NOx. Also, the authors do not address whether biogenic VOCs have any impact on the VOC budget. In addition, ozone is co-produced with formaldehyde in high-NOx conditions, and thus as VOC oxidation increases as NOx levels go down, it is possible that increased HCHO production and even a lengthening of the HCHO lifetime buffers the signal from any VOC emissions reductions.

I do not believe that the analysis presented by the authors is convincing and novel in its current form. I would suggest that the authors improve their analysis and resubmit at a later time. The manuscript could also benefit from an English language edit.

**Minor comments:**

Line 54 - Schroeder et al., 2017 (http://dx.doi.org/10.1002/2017JD026781) showed that there is a large range of ambiguity in CH2O/NO2, and that inhomogeneities in vertical mixing degrade the ability of column HCHO/NO2 to classify surface ozone sensitivity. They also show that ozone sensitivity on exceedance days may be different than non-exceedance days. The authors need to address these points.

Line 59 – According to Zheng et al., 2018a, the change in VOC source categories is large. Does this have an impact on HCHO?

Line 80 – I am unable to access this link - (http://106.37.208.233:20035/).

Line 81 – Can you explain what you mean by 'unified to the reference state'?

Line 83 – Do you mean the daily average?

Line 98 – Do you mean 'chemical transport model'?

Line 105 – Why not just use OMI through 2021?

Line 110 – Can you just clarify whether OMI and TROPOMI use the same NO2 a priori?

Figure 1a – Just show the ozone season, May-July. Clearly from panel b) there is a strong seasonality, and we are more interested in increases in ozone during that time that annual changes. On panel b, show a trend just through May-July vs. Dec-Feb.

Figure 3 – Why is this data now only April – September? Be consistent with your time periods. Panel b) – Is this just 13-14 local time?

Line 191-192 - I do not understand the reference to Pusede and Cohen, 2012. Nowhere in that paper do they show ozone as a function of HCHO/NO2.

Given that Wang et al., 2022 show different trends in Beijing, Shanghai, and Guangzhou, the authors should take the approach of Jon and Holloway (2020) and separate their analysis by city (i.e. Fig. 1b).

Could the authors please explain why they averaged the data into half month increments.

Line 195 - I think it would be more useful to show Figure S2 in place of Figure 3. I am not sure how it is useful to average all the sites together when there are strong regional differences.

Line 201 – You say "This varied response is also reflected in different thresholds for other photochemical indicators ... or patterns of O3 isopleths...". Please be specific about what these other studies say and how it fits with your results, right now this sentence is too speculative to be useful.

Line 202 – You should have the information to actually discuss changes in temperature and radiation across these regions and be a little more specific about whether the could impact HCHO/NO2.

Line 207 – Give the previously reported values.

Line 216 – Given that you have shown strong regional differences, I do not think it makes sense to show Fig. S3 at all, and move forward only with regionally-specific analysis.

Figure S6 – It is very confusing to have the timeseries split between the supplement and main text. I don't see why putting Fig S6 into Figure 5 and 6 would be problematic.

Figure 5b and 6b – Why are you showing annual trends when your ozone analysis is April – September? Stick to analyzing the same time period throughout.

Line 251 - I don't understand the statement "Based on the non-linear relationship of O3-NOx-VOC, the changes in ozone precursors is the direct factor affecting ozone level." Are you trying to argue that other factors like meteorology or changes in PM are not the cause? If so, this statement is not convincing as there has been no discussion of these other factors.

Figure S7 – Show the same thing for HCHO.

Figure 6b – Are those trends really statistically significant? Also you need to look at just April-September.

Line 269 – Could it not also be an indicator that as urban ozone titration decreases, and OH increases due to the reduced NOx level, that oxidation of VOCs increases?

Line 271 – The authors have not discussed the importance of biogenic isoprene to the VOC budget.

Line 273 – Show us a temperature timeseries.

Line 311 – The authors are assuming that HCHO columns directly relate to VOC emissions, while changes in chemistry due to NOx reductions could have a large impact (see my above comment). Thus I do not think you can state that the increase in HCHO is due to increased VOC emissions without further evidence. The authors need to explain how the "vigorous development of stall businesses" could cause this increase.

Line 325 – The authors haven't discussed PM2.5 at all to support this statement.

Figure 8 – I don't understand the purpose of this figure.

---

## Author Response (AR1)

Dear Editor,

We are sincerely happy to have the chance to submit the revised manuscript and are very grateful for your help throughout the process. We also owe sincere thanks to both reviewers' valuable suggestions, which we believe have helped to improve the quality of the manuscript significantly. Detailed responses to reviewers' comments were listed below, and the changes were made in the revised manuscript based on both reviewers' comments accordingly. All changes are marked in blue.

**Response to Referee #1:**

*This study combined satellite observations of NO₂ and HCHO with surface air quality measurements over China to characterize ozone formation sensitivities and its long-term changes. A methodology largely consistent with Jin et al. (10.1021/acs.est.9b07785, 2020) were applied to harmonize long-term data from OMI and TROPOMI, and to diagnose the ozone sensitivity regimes. Then changes of satellite NO₂ and HCHO were used to interpret the recent ozone increases, which is mainly attributed to NO₂ reductions while VOC changed little in the context of the VOC-limited regime. Two examples showing ozone responses during COVID-19 lockdown were discussed to complementally support their arguments.*

*While the topic is definitely within the scope of ACP and the methodology & results are clearly written, easy to follow and overall sound, I have moderate reservations regarding publishing the manuscript at its present form. The main issue is about its novelty which I will outline later. I will support the publication of this paper, if the following concerns can be adequately addressed.*

**Response:**

We thank the comments and suggestions from the reviewers, which help to improve the manuscript considerably. The response and changes are listed below.

**Major points:**

**1.** *Novelty. The whole idea of using satellite observations of HCHO and NO₂ to identify an indicator of ozone formation regime and to interpret ozone changes, as the authors also described, has been proposed and applied for ~two decades. In particular, I can easily list two recent ACP papers that essentially used the same idea, almost the same data and processing, applied to the same domain (China) and overall consistent time period (since 2010s), and came up with consistent conclusions: Wang et al. (10.5194/acp-21-7253-2021, 2021) and Li et al. (10.5194/acp-21-15631-2021, 2021). I merely found substantially novel/additional insights from this manuscript relative to these two papers. In the next round the review, the authors should highlight the unique points in their manuscript relative to these two papers to justify that their paper is substantially novel.*

**Response: Accepted.**

The uniqueness of this manuscript relative to these two papers includes the following aspects. Firstly, higher resolution TROPOMI data is used to improve the accuracy of

estimating the HCHO/NO$_2$ thresholds in different regions of China. Secondly, some methodology details for establishing the relationship between O$_3$ and HCHO/NO$_2$ are different from those mentioned by the reviewer. For example, the hourly ozone concentration is directly used; the range of the transition regime is defined by the slope of the curve. Thirdly, O$_3$ sensitivity distributions in China are updated to 2021 using these thresholds and the reasons why ozone concentrations increased in recent years combined with long-term variation trends of ozone precursors and short-term change cases are explored. Furthermore, optimal VOC/NO$_x$ reduction ratios for three typical cities are provided.

We added and emphasized in Line 55-59 of the revised manuscript. "In addition, most previous studies utilized the OMI satellite HCHO/NO$_2$ (Li et al., 2021; Wang et al., 2021). The newly launched TROPOMI aboard Sentinel-5P provides a new perspective to characterize the chemistry of surface O$_3$ at finer spatial scales (Veefkind et al., 2012). Therefore, for a more accurate diagnosis of the ozone-precursor sensitivity across China, the transitional regime range of HCHO/NO$_2$ using the new generation of satellites requires further in-depth investigation."

We added the relevant discussion in Line 345-349 "Based on the evaluation of the nonlinearity of O$_3$-NO$_x$-VOC chemistry captured by space-based HCHO/NO$_2$, this study derives thresholds marking the transitions between chemical regimes in different regions of China and diagnoses the current spatial distribution of O$_3$ production regimes. The higher resolution TROPOMI satellite data enables a better match between the location of ground-based O$_3$ monitoring stations and the grid of satellite data, thus allowing a more accurate derivation of HCHO/NO$_2$ thresholds and reflection of spatial variations in ozone-NO$_x$-VOC sensitivity."

*2. Potentially unnecessary data processing. The idea of "harmonization" of OMI and TROPOMI data might be borrowed from Jin et al. (10.1021/acs.est.9b07785, 2020), but it is odd to be used here for 2013-2021, which is already fully covered by OMI. The "harmonization" process will introduce potentially more uncertainties, especially considering the short time period (thus the climatological differences between the two sensors and the adjustments based on that will be more contaminated by meteorological anomalies). Furthermore, I do not see any unique insights/interpretations that are only available at <20 km spatial scale in the manuscript. If the "harmonized data" is still used in the revision, the authors should evaluate how different the results become if only using OMI data, and justify that these differences are strong and due to the unique information from TROPOMI.*

**Response: Accepted.**

To study the long-term changes in NO$_2$ and HCHO, we adjusted to use OMI data for 2013-2020 and TROPOMI data for 2021, since the OMI data retrieved by the QA4ECV project are available until December 2020. We converted TROPOMI data for 2021 based on a comparison of OMI and TROPOMI data for 2019-2020 to avoid differences in any instrumental offset.

We keep using TROPOMI data to derive the HCHO/NO$_2$ threshold values and to diagnose the current ozone-NO$_x$-VOC sensitivity for several reasons. First, the higher resolution allows for more accurate identification of the spatial distribution of ozone-NO$_x$-VOC sensitivity, such as detailed spatial patterns within urban cores, which facilitates targeted emission control. Second, considering that most air monitoring stations are located in urban areas, the high resolution allows for a more exact match between the location of monitoring stations and the grid of satellite data, thus enabling a more accurate derivation of HCHO/NO$_2$ thresholds.

We revised the last paragraph in section 2.2 as follows "OMI data with longer time horizons (2013-2020) are used to study the long-term changes in NO$_2$ and HCHO through their monthly averages and track changes in emissions of NO$_x$ and VOCs. Since the OMI data retrieved by the QA4ECV project are available until December 2020, the data for 2021 are obtained by converting the TROPOMI data, which is based on a comparison of OMI and TROPOMI monthly data for 2019-2020 to avoid differences in any instrumental offset."

We added a sentence in Line 113 "The high spatial resolution TROPOMI data is used to derive the HCHO/NO$_2$ threshold values marking transitions in O$_3$ formation regimes and to diagnose the current ozone-NO$_x$-VOC sensitivity in China." in section 2.3.

We added sentences in the Introduction "In addition, most previous studies utilized the OMI satellite HCHO/NO$_2$ (Li et al., 2021; Wang et al., 2021). The newly launched TROPOMI aboard Sentinel-5P provides a new perspective to characterize the chemistry of surface O$_3$ at finer spatial scales (Veefkind et al., 2012). Therefore, for a more accurate diagnosis of the ozone-precursor sensitivity across China, the transitional regime range of HCHO/NO$_2$ using the new generation of satellites requires further in-depth investigation." in Line 55-59.

We added a sentence in Line 113 to emphasize the role of TROPOMI data "The high spatial resolution TROPOMI data is used to derive the HCHO/NO$_2$ threshold values marking transitions in O$_3$ formation regimes and to diagnose the current ozone-NO$_x$-VOC sensitivity in China."

***3.*** *The COVID-19 analysis. Section 3.4 is confusing to me. First, the main topic is to discuss long-term changes (and maybe relevance with emission regulations), I believe this short-term ozone responses to $NO_2$ and HCHO do not support the long-term analysis before. Second, the Chinese lockdown is during February and March, 2020. The authors selected April for Beijing, and May for Chengdu, why? Indeed, both $NO_2$ and HCHO do not show the "COVID-typical" reductions to me. Third, can the current analysis for one month in three years, each year with their unique meteorological conditions/variations, really support the attribution of these ozone responses to be driven by emissions? Including more years that potentially envelope possible meteorological variabilities seem more reasonable to me.*

**Response: Accepted.**

This section builds on the long-term relationships between changes in ozone and its precursors discussed in the previous section and verifies the $O_3$ sensitivity to large changes in emissions and ozone concentrations in cities in the short term through case studies. This study focuses on the $O_3$ sensitivity from April to September, the period when ozone exceedances are most likely to occur. In April 2020, Beijing is still under strict restrictions because of epidemic control. Therefore, $NO_2$ concentration is significantly lower than those in 2019 and 2021, which is caused by the still restricted anthropogenic activities rather than by the air pollution control actions. The significant increase in HCHO concentrations in Chengdu in May 2020 implies an increase in VOC emissions. Song et al. (2021) also reported an increase in VOC concentrations. It could be explained by the vigorous development of stall business in Chengdu at that time in order to the resumption of work and production.

In both cases, emissions and meteorological conditions (temperature and relative humidity) jointly contributed to the increase in ozone concentrations compared to the previous and the following year, and the effects of these two were not distinguished. From the results, the decrease in $NO_x$ and the increase in VOCs, as well as the unfavorable meteorological conditions, contributed to the ozone pollution event, which verifies the results of the previous sections of this study.

We moved this section to 3.3.2 and combined it with the discussion of long-term changes as Section 3.3, Effects of Ozone Precursors Variations on Ozone. We revised the first paragraph of this section in Line 297 "The outbreak of the COVID-19 pandemic produced previously unseen societal impacts in China. The measures during the plague prevention and work resumption resulted in changes in VOC and $NO_x$ emissions in 2020 compared to normal years (Le et al., 2020; Pei et al., 2022), and these changes were not synchronized across Chinese cities. The ozone pollution that occurred in Beijing and Chengdu during

this period is used as natural experiments to evaluate surface $O_3$ responses to apparent emission variations in order to validate the conclusions above."

We added a paragraph in Line 322 "In these two cases in Beijing and Chengdu, meteorological conditions also contributed to the increase in ozone concentrations. As shown in Figure 7, compared to the previous and subsequent years, there is only a slight increase in temperature; relative humidity decreases, but remains high in Chengdu (over 60%); and wind speed is not the lowest in the three years. Although the effects of emissions and meteorological conditions are difficult to distinguish quantitatively, it reflects the negative effect of $NO_x$ reduction alone or VOC increase on $O_3$ pollution in urban cities."

*4. Strong arguments about potential PM effects. The authors concluded that "Our study highlights that the root cause of ozone increase in major regions is the significant reduction of $NO_x$ alone without effective control of VOC and not the concurrent decreases in the $PM_{2.5}$ level as suggested in previous studies". However, their results cannot support this argument. $PM_{2.5}$ and $NO_x$ decrease simultaneously during the investigated period, therefore ozone increases are also associated with $PM_{2.5}$ reductions. Whether the chemical regime is $NO_x$-limited or $NO_x$-saturated does not rule out the PM effects, since uptake of HO2 will affect both regimes according to Li et al. (2019). If the authors would like to retain their strong arguments that PM is not affecting the ozone production, they will need to validate that ozone at similar HCHO and NO2 level (e.g. bins in Figure 3a, with meteorological effects also minimized/normalized) stays the same over time.*

**Response: Accepted.**

Some studies suggested that concurrent decreases in the $PM_{2.5}$ level may be a potential driver of ozone increases in China, which may increase solar radiation and weaken the aerosol sink of $HO_2$ radicals and thus stimulate ozone production. However, the summertime MDA8 $O_3$ enhancement due to changes in emissions and $PM_{2.5}$ levels in NCP during 2013-2017 estimated by Li et al. (1 ppb year$^{-1}$) is insufficient to explain the observed trend (3.3 ppb year$^{-1}$) (Lu et al., 2020), which indicates that the effect of $PM_{2.5}$ is not the essential cause of the deterioration of ozone pollution.

The results in this study show that it is the variation of the ozone precursors VOC and $NO_x$ that is responsible for the ozone concentrations. Our previous study also shows that $O_3$ concentrations in Beijing have declined in recent years in parallel with the significant decline in $PM_{2.5}$ concentrations, suggesting that the appropriate emission reduction ratio of VOC to $NO_x$ can control both $PM_{2.5}$ and $O_3$ pollution. A simple emphasis on "decreasing

PM$_{2.5}$ leads to increasing ozone concentrations" could mislead the government to make the wrong policy decisions.

We have revised the title of Section 4 to "Discussion", and the following relevant discussion has been added to Line 368-373 of the revised manuscript. "Some studies suggested that the significant decrease in PM$_{2.5}$ is the most crucial factor contributing to the O$_3$ increment in China (Li et al., 2019), but the idea of the impact of HO$_2$ uptake is controversial (Tan et al., 2020). The summertime MDA8 O$_3$ enhancement due to changes in PM$_{2.5}$ levels in NCP during 2013-2017 estimated by Li et al. (1 ppb year$^{-1}$) is insufficient to explain the observed trend (3.3 ppb year$^{-1}$) (Lu et al., 2020), which indicates that the effect of PM$_{2.5}$ is not the essential cause of worsening ozone pollution. The simultaneous decline in O$_3$ and PM$_{2.5}$ concentrations in Beijing in recent years (Ren et al., 2021) suggests that appropriate VOC and NO$_x$ emission reductions can control both PM$_{2.5}$ and O$_3$ pollution."

**Minor points:**

**1.** *The "x" in "NO$_x$" should be a subscript.*

**Response: Accepted.**

The "NOx" has been revised to "NO$_x$" in the revised manuscript.

**2.** *Line 63-64: "highly controversial" due to one paper finding inconsistency over one site?*

**Response: Accepted.**

Although Tan et al.'s study is over one site, they suggest that HO$_2$ uptake on aerosol did not play a role in determining peroxy radical concentrations, completely denying Li et al.'s point. In addition, Lu et al. discovered that summertime MDA8 O$_3$ enhancement due to changes in PM$_{2.5}$ levels in NCP during 2013-2017 estimated by Li et al. (~1 ppb year$^{-1}$) is insufficient to explain the observed trend (3.3 ppb year$^{-1}$).

For precise expression, this sentence has been revised to "However, this idea remains controversial (Tan et al., 2020)." in Line 65 in the revised manuscript. We added relevant discussions in section 4.

***3.*** *Beijing and Chengdu are selected to look at COVID-19 effects (Figure 7), and Beijing, Chengdu and Guangzhou are used to investigate optimal emission regulation (Figure 8). Why are these cities selected? Can they represent other cities?*

**Response: Accepted.**

Ozone pollution occurring in Beijing and Chengdu during the plague prevention and the work resumption is used as natural experiments to evaluate surface $O_3$ responses to apparent emission variations. The purpose of this section is to verify the above conclusions with these two cases. We changed Line 299 "The ozone pollution that occurred in Beijing and Chengdu during this period is used as natural experiments to evaluate surface $O_3$ responses to apparent emission variations in order to validate the conclusions above."

At the end of this study, we studied the VOC/$NO_x$ reduction ratios in three cities, Beijing in the North China Plain, Guangzhou in Pearl River Delta, and Chengdu in Sichuan Basin. These three developed cities, dominated by the VOC-limited regime, are used as examples to specifically provide preliminary solutions for the synergistic reduction of VOC and $NO_x$ and do not represent all cities in China. We revised in Line 334 "This approach is used to provide preliminary scenarios for synergistic VOC and $NO_x$ reduction in three developed cities, Beijing in the NCP, Guangzhou in the PRD, and Chengdu in the SCB, which are all dominated by the VOC-limited regime."

***4.*** *Line 81: The "reference state" was at 273 K before September 2018, and at 298 K afterwards. Is this factor considered? Should be included in the introduction.*

**Response: Accepted.**

Ozone and $NO_2$ measurements are reported at the standard atmospheric condition (273.15K, 1013.25 hPa) before September 2018 and at 298.15K conditions afterward, which has been considered in the original manuscript. All mass concentrations were unified to the reference state (298.15 K, 1013.25 hPa).

To avoid misunderstandings, this sentence has been revised to "$O_3$ and $NO_2$ measurements were reported in μg m$^{-3}$ and the mass concentrations for all years were unified to the reference state (298.15 K, 1013.25 hPa)." in Line 84 in the revised manuscript.

***5.*** *Line 97-103: please provide more detailed introduction of the data. How the re-gridding is done? What is the temporal resolution of data used?*

**Response: Accepted.**

We calculated HCHO/NO$_2$ using TROPOMI data with the spatial resolution of 0.01° from 2019-2021 to study the sensitivity of ozone generation in the revised manuscript, thus no resampling is involved.

The temporal resolution of the raw satellite data is per day. We added this information in Line 96 "The high spatial resolution (24 × 13 km$^2$ for OMI and 5 × 3.5 km$^2$ for TROPOMI) allows for the observation of fine details of atmospheric parameters. They provide daily global observations, and the overpass time (13:40–13:50 and 13:30 local time) is well suited to detect the O$_3$ formation sensitivity."

***6.*** *Line 118: Please provide a map of the 9 regions for people unfamiliar with geography of China.*

**Response: Accepted.**

The map of the 9 regions of China has been provided in Figure S2 in the Supplement.

[Figure]

Figure S2. Map of the nine regions into which China is divided in this study.

***7.*** *Figure 3a: Some isopleth lines will greatly help guide the audience.*

**Response:**

We tried to draw some isopleth lines on the basis of Figure 3a, however, since the data here are entirely from observations and affected by many factors, it is difficult to obtain ideal isopleth lines. Nonetheless, it is evident from the present figure that satellite-based HCHO/NO$_2$ captures the well-established nonlinearities in O$_3$ chemistry.

*8. Figures 2/4/5/6: Are the annual maps really necessary? Maybe one map for each phase (2013, 2019, 2021) will be enough?*

**Response: Accepted.**

Our purpose is to show the spatial distribution of $O_3$ concentrations each year in detail in Figure 2, including the expansion and contraction of high-value areas.

We explored the distribution of ozone sensitivity only for 2019-2021 using TROPOMI data in the revised manuscript, as shown in Figure 4.

[Figure]

Figure 4: Ozone sensitivity classification over China from April to September 2019–2021 using different HCHO/$NO_2$ thresholds in different regions. Only polluted regions are displayed (defined as average TROPOMI $NO_2$ columns higher than $1.0 \times 10^{15}$ molecules/cm$^2$).

We have replaced Figures 5a and 6a to show only 2013, 2017, and 2020, based on the utilization of OMI data in the revised manuscript.

(a) Satellite NO$_2$ (molec/cm$^2$)

Satellite NO$_2$ in eastern China

Annual trend (10$^{15}$ molec. cm$^{-2}$ yr$^{-1}$)
-0.10 (-2.9%),   p<0.01

Satellite NO$_2$ in NCP

Annual trend (10$^{15}$ molec. cm$^{-2}$ yr$^{-1}$)
-0.32 (-4.5%), p<0.01

Monthly verticle column (10$^{15}$ molec./cm$^2$)

Figure 5: (a) Maps of average satellite-based NO$_2$ columns over China from April to September, and (b) monthly mean NO$_2$ columns averaged over eastern China and North China Plain in April-September. Gray shading: mean value ± 50% standard deviation across all grids for each month. Solid line: the linear fitted curve. Inset: absolute annual linear trend and percentage of annual trend (% per year, the linear trend divided by the 2013 mean values).

[Figure]

Figure 6: Same as Figure 5 but for satellite-based HCHO columns.

**9.** *Figure 7: Without e.g. a modeling framework to isolate each contribution, just listing all the monthly-average numbers of these parameters cannot support the discussion in Section 3.4*

**Response: Accepted.**

As in the response to major point 3, in both cases, emissions and meteorological conditions (temperature and relative humidity) together contributed to the increase in ozone concentrations over the previous year and the following year. The effects of these two factors were not quantitatively distinguished. From the results, the decrease in $NO_x$ and the increase in VOCs, together with unfavorable meteorological conditions, contributed to the ozone pollution events, which validates the results of the previous sections regarding the relationship between ozone and its precursors.

We added related discussions in Line 322-326 "In these two cases in Beijing and Chengdu, meteorological conditions also contributed to the increase in ozone concentrations. As shown in Figure 7, compared to the previous and subsequent years, there is only a slight increase in temperature; relative humidity decreases, but remains high in Chengdu (over 60%); and wind speed is not the lowest in the three years. Although the effects of emissions and meteorological conditions are difficult to distinguish quantitatively, it

reflects the negative effect of NO$_x$ reduction alone or VOC increase on O$_3$ pollution in urban cities."

*10. Figure 8 and its relevant discussion: It is unusual to introduce more results in the last section. Please consider re-organize.*

**Response: Accepted.**

We restructured the manuscript by merging the original Section 3.4 into 3.3 and moving Figure 8 and its relevant discussion to Section 3.4 in the revised manuscript.

**Response to Referee #2:**

**Major comments:**

*1. I am concerned that the authors suggest the need to shift from a focus on $NO_x$ reduction to VOC reduction. Once $NO_x$ reductions cease to result in increased rates of ozone production due to the non-linear response of ozone to $NO_x$, ozone levels will again start to decrease. Thus continued NO x reductions will eventually reduce ozone even without reducing VOCs. Certainly there are plenty of reasons to reduce VOCs, but continued $NO_x$ reductions will eventually reduce ozone and efforts should not move away from reducing $NO_x$. The authors could consider calculating at what point that might start to occur. This paper, Wang et al., 2022 - https://doi.org/10.5194/acp-22-8935-2022 – used box modeling to show that cities transitioned from VOC-limited to a transition regime, and concluded that China needs concurrent reductions in both VOC and $NO_x$ reductions. The authors need to address the findings of this paper, and differentiate their analysis, which is similar in assessing the long-term trends in ozone.*

**Response: Accepted.**

Indeed, ozone is ultimately formed from the combination reaction of atomic oxygen ($O_3P$) and molecular oxygen ($O_2$). In the troposphere, photolysis of $NO_2$ at wavelengths < 424 nm becomes the primary source of $O_3P$ atoms and prompts $O_3$ formation. Therefore, a deep reduction of $NO_x$ will eventually reduce the $O_3$ concentration. However, the presence of VOC and OH radical lead to $O_3$ accumulation and the non-linear dependence of $O_3$ production on its precursors, i.e., $NO_X$ and VOCs.

Our results show that the $NO_x$-limited regime exists in a large area of China currently, where $NO_x$ reduction is effective in controlling ozone pollution. However, at present, key city clusters and urban areas are dominated by VOC-limited and transitional regimes. If the scheme of no VOC reduction and $NO_x$ reduction only is adopted, the $O_3$ concentration cannot be effectively reduced until $NO_x$ is reduced to a very low level to lead the $O_3$ formation regime to shift to $NO_x$-limited. This would take a long time and be very costly, which is unacceptable to the government and the public. In contrast, a strong effort to reduce VOCs and increase the VOCs/$NO_x$ reduction ratio is the immediate and effective way to reduce $O_3$ concentrations.

The results of Wang et al. (2022) show that VOC reduction would significantly decrease $O_3$, while $NO_x$ reduction would only slightly decrease $O_3$ at two urban sites in Beijing and Shanghai from June to August after 2019. This is consistent with our findings from April to September in Beijing and Shanghai that VOC reduction is more effective than $NO_x$

reduction for controlling $O_3$ pollution. The difference in study months may lead to $O_3$ production being more VOC-limited in our results. In addition, we provide the spatial distribution of ozone sensitivity in China, and different regions should adopt different emission reduction strategies.

Considering your comments, the following relevant description has been added to Line 374−381 in section 4 of the revised manuscript. "Currently, the VOC-limited regime exists in most major urban clusters in China, which are also areas with high ozone levels. The exploration of the causes of ozone rise in China illustrates the current need for VOC control, which is consistent with the findings of previous studies on major cities and city clusters (Wang et al., 2022; Yang et al., 2019). Deep $NO_x$ reductions will eventually reduce $O_3$ concentrations to lower levels. However, if only $NO_x$ is currently reduced without VOC reductions, $O_3$ concentrations will not be effectively reduced until $NO_x$ is reduced to very low levels, leading to a shift in $O_3$ formation mechanisms toward $NO_x$-limited. This would take a long time and be very costly. In contrast, synergistic control of $NO_x$ and VOC, with a focus on VOC control, is currently a direct and effective approach to alleviate ozone pollution in large cities."

*2. The authors do not provide a convincing argument that the increase in satellite HCHO is due to a lack of emissions reductions. According to Zheng et al., 2018a referenced by the authors, the VOC composition has changed significantly with reductions in residential and transportation VOCs but increases in industrial and solvent VOCs. The authors need to discuss the impact of the change in VOC composition, particularly any shift in yield of HCHO from the change in VOC mixture and the reduction in $NO_x$. Also, the authors do not address whether biogenic VOCs have any impact on the VOC budget. In addition, ozone is co-produced with formaldehyde in high-$NO_x$ conditions, and thus as VOC oxidation increases as $NO_x$ levels go down, it is possible that increased HCHO production and even a lengthening of the HCHO lifetime buffers the signal from any VOC emissions reductions.*

**Response: Accepted.**

The change in VOC source categories leads to different changes in emissions of different VOC species. Although the HCHO yields of different VOC species differ, the changes in satellite HCHO can roughly indicate changes in total VOC emissions, which has been applied in several previous studies (Li et al., 2020; Shen et al., 2019; Zhu et al., 2014).

Our results show that there is no significant overall decrease in satellite HCHO concentrations in most of eastern China since 2013, which is similar to the estimated trends of anthropogenic VOC and biogenic VOC emissions in China (Zheng et al., 2018;

Simayi et al., 2022; Li et al., 2020). Biogenic VOC increases slightly, while anthropogenic VOC emissions in urban areas in China are much higher than biogenic VOC emissions. Therefore, it is the lack of anthropogenic VOC emission reduction that has caused no significant decrease in HCHO.

In addition to what the reviewer mentioned, the HCHO yield of some VOCs (e.g., isoprene) decreases with decreasing $NO_x$ concentrations. Overall, the long-term changes in HCHO are driven by several factors, such as anthropogenic and biogenic emissions, OH abundance, and $NO_x$ concentrations, which deserve further investigation in future studies using more adequate observational data. Most relevant to our study is that the decline in HCHO is much less than that of $NO_2$. Therefore, this reflects a smaller overall change in VOC than $NO_x$.

Considering your comments, we added a sentence in Line 269 "The changes in satellite HCHO can roughly indicate changes in VOC emissions, which has been applied in several previous studies (Li et al., 2020; Shen et al., 2019; Zhu et al., 2014)." We added the relevant discussion in Line 350 of the revised manuscript. "However, the use of satellite $HCHO/NO_2$ for quantitative diagnosis of ozone-$NO_x$-VOC sensitivity is influenced by the following uncertainties and requires further investigation." "Second, the relationship between satellite-based HCHO and surface VOC reactivity is not fully understood. There are differences in HCHO yields for different classes of VOCs (Shen et al., 2019). Biogenic VOC emissions also have an influence. Since HCHO is a weaker UV-visible absorber than $NO_2$, satellite retrieval of HCHO is more error-prone. These factors may limit their usefulness in detecting HCHO from local sources of anthropogenic VOCs."

*3. I do not believe that the analysis presented by the authors is convincing and novel in its current form. I would suggest that the authors improve their analysis and resubmit at a later time. The manuscript could also benefit from an English language edit.*

**Response:**

We thank the comments and suggestions from the reviewers, which help to improve the manuscript considerably. The response and changes are listed. We have also carefully checked and improved the English writing in the revised manuscript.

**Minor comments:**

*1. Line 54 - Schroeder et al., 2017 (http://dx.doi.org/10.1002/2017JD026781) showed that there is a large range of ambiguity in $CH_2O/NO_2$, and that inhomogeneities in vertical*

*mixing degrade the ability of column HCHO/NO$_2$ to classify surface ozone sensitivity. They also show that ozone sensitivity on exceedance days may be different than non-exceedance days. The authors need to address these points.*

**Response: Accepted.**

HCHO/NO$_2$ can be used to indicate O$_3$ sensitivity, but there are some uncertainties, such as the transition range and span of regional differences, inhomogeneities in vertical mixing in the lower troposphere, and differences in exceedance days and non-exceedance days.

This study derives the range of HCHO/NO$_2$ marking transitions in O$_3$ formation regimes entirely from observations, which vary regionally. Based on this result, the spatial distribution of O$_3$ production sensitivities for April-September in China is provided. This overall result avoids some uncertainties and can give a great reference to the government to develop emission reduction strategies.

Considering your comments, the following relevant description has been added to Line 350 of the revised manuscript. "However, the use of satellite HCHO/NO$_2$ for quantitative diagnosis of ozone-NO$_x$-VOC sensitivity is influenced by the following uncertainties and requires further investigation. First, the thresholds derived from the observation-based approach may be affected by biases in the satellite retrieval algorithm. Satellite instruments measure vertically integrated column densities, and the inhomogeneity of the vertical distribution might impose negative influences (Schroeder et al., 2017). Second, the relationship between satellite-based HCHO and surface VOC reactivity is not fully understood. There are differences in HCHO yields for different classes of VOCs (Shen et al., 2019). Biogenic VOC emissions also have an influence. Since HCHO is a weaker UV-visible absorber than NO$_2$, satellite retrieval of HCHO is more error-prone. These factors may limit their usefulness in detecting HCHO from local sources of anthropogenic VOCs."

*2. Line 59 – According to Zheng et al., 2018a, the change in VOC source categories is large. Does this have an impact on HCHO?*

**Response: Accepted.**

As in the response to the major comment 2, the HCHO yields of different VOC species differ. The change in VOC source categories leads to different changes in emissions of different VOC species, which are difficult to represent by the HCHO concentration. In this study, the changes in satellite HCHO are used to roughly indicate changes in total VOC emissions, which has been applied in several previous studies (Li et al., 2020; Shen et al., 2019; Zhu et al., 2014).

Considering your comments, the following relevant description has been added to Line 269, "The changes in satellite HCHO can roughly indicate changes in VOC emissions, which has been applied in several previous studies (Li et al., 2020; Shen et al., 2019; Zhu et al., 2014)." and to Line 350 of the revised manuscript "However, the use of satellite HCHO/NO$_2$ for quantitative diagnosis of ozone-NO$_x$-VOC sensitivity is influenced by the following uncertainties and requires further investigation." "Second, the relationship between satellite-based HCHO and surface VOC reactivity is not fully understood. There are differences in HCHO yields for different classes of VOCs (Shen et al., 2019) …These factors may limit their usefulness in detecting HCHO from local sources of anthropogenic VOCs."

**3.** *Line 80 – I am unable to access this link - (http://106.37.208.233:20035/).*

**Response: Accepted.**

The link to the National Urban Air Quality Real-Time Publishing Platform has been updated to https://air.cnemc.cn:18007/.

**4.** *Line 81 – Can you explain what you mean by 'unified to the reference state'?*

**Response: Accepted.**

Ozone and NO$_2$ measurements are reported at the standard atmospheric condition (273.15K, 1013.25 hPa) before September 2018 and at 298.15K conditions afterward. All mass concentrations were unified to the reference state (298.15 K, 1013.25 hPa).

To avoid misunderstandings, this sentence has been revised to "O$_3$ and NO$_2$ measurements were reported in μg m$^{-3}$ and the mass concentrations for all years were unified to the reference state (298.15 K, 1013.25 hPa)." in Line 84 in the revised manuscript.

**5.** *Line 83 – Do you mean the daily average?*

**Response: Accepted.**

The daily NO$_2$ mean concentration of the city is the average daily NO$_2$ at all sites, and the O$_3$-MDA8 of the city is the average O$_3$-MDA8 of all sites.

In order to make the expression clearer, this sentence has been revised to "The average of daily NO$_2$ and the daily maximum eight-hour average (MDA8) O$_3$ concentration for all

national-controlled sites in a city was regarded as the city's daily $NO_2$ and $O_3$ levels." in Line 86 of the revised manuscript.

*6. Line 98 – Do you mean 'chemical transport model'?*

**Response: Accepted.**

We are sorry for the mistake. It has been revised to "chemical transport model" in Line 102 of the revised manuscript.

**7.** *Line 105 – Why not just use OMI through 2021?*

**Response: Accepted.**

We have made the following adjustments. OMI data for 2013-2020 and TROPOMI data for 2021 are used to study long-term changes in $NO_2$ and HCHO due to the availability of OMI data retrieved by the QA4ECV project through December 2020. The data for 2021 is obtained by converting the TROPOMI data, which is based on a comparison of OMI and TROPOMI data for 2019-2020 to account for differences in any instrumental offset. The higher spatial resolution TROPOMI data is used to derive the HCHO/$NO_2$ threshold values and to diagnose the current ozone-$NO_x$-VOC sensitivity in China.

The following relevant descriptions replaced the previous ones in Line 108 of the revised manuscript "OMI data with longer time horizons (2013-2020) are used to study the long-term changes in $NO_2$ and HCHO through their monthly averages and track changes in emissions of $NO_x$ and VOCs. Since the OMI data retrieved by the QA4ECV project are available until December 2020, the data for 2021 are obtained by converting the TROPOMI data, which is based on a comparison of OMI and TROPOMI monthly data for 2019-2020 to avoid differences in any instrumental offset."

**8.** *Line 110 – Can you just clarify whether OMI and TROPOMI use the same $NO_2$ a priori?*

**Response: Accepted.**

As in the response to comment 7, we have replaced the relevant methods in Line 109 of the revised manuscript "Since the OMI data retrieved by the QA4ECV project are available until December 2020, the data for 2021 are obtained by converting the TROPOMI data, which is based on a comparison of OMI and TROPOMI monthly data for 2019-2020 to account for differences in any instrumental offset."

***9.*** *Figure 1a – Just show the ozone season, May-July. Clearly from panel b) there is a strong seasonality, and we are more interested in increases in ozone during that time that annual changes. On panel b, show a trend just through May-July vs. Dec-Feb.*

**Response: Accepted.**

The three metrics shown in Figure 1a following the definitions in World Health Organization global air quality guidelines and Chinese ambient air quality standards are common indicators for evaluating ozone levels. Considering that this study focused on April-September, Figure 1b is changed to only show the monthly average MDA8 in April-September each year.

[Figure]

Figure 1: Surface ozone concentrations in China. (a) MDA8-99th, MDA8-90th, MMA6-MDA8, and (b) monthly average (black dots, left axis) and anomaly (circle, right axis) of MDA8 ozone concentrations averaged from all 367 cities with monitoring sites in 2015–2021. Gray shading: mean value ± standard deviation across all cities for each month. Solid line: linear fitted curve. (c) Ozone diurnal cycles from April to September.

***10.*** *Figure 3 – Why is this data now only April – September? Be consistent with your time periods. Panel b) – Is this just 13-14 local time?*

**Response: Accepted.**

We are primarily concerned with the ozone sensitivity from April to September and therefore use the April-September data when deriving the $HCHO/NO_2$ thresholds in Figure 3.

The ozone concentration in Fig. 3b is at 13:00 to 14:00 local time, which is derived on the basis of Fig. 3a. We added "13:00 to 14:00" to the title of Fig. 3b, and indicated in the related introduction that the data in Figure 3b is "from Figure 3a" in Line 190.

**11.** *Line 191-192 – I do not understand the reference to Pusede and Cohen, 2012. Nowhere in that paper do they show ozone as a function of HCHO/NO₂.*

*Given that Wang et al., 2022 show different trends in Beijing, Shanghai, and Guangzhou, the authors should take the approach of Jon and Holloway (2020) and separate their analysis by city (i.e. Fig. 1b).*

*Could the authors please explain why they averaged the data into half month increments.*

**Response: Accepted.**

We have removed the reference to this paper in the revised manuscript.

We investigate the relationship between $O_3$ concentration and TROPOMI HCHO/NO₂ for each of the nine regions in China. For each city, the number of air quality monitoring stations is limited, and therefore the amount of data is insufficient to support the establishment of such a fitted relationship. We rewrote the sentences in Line 233 of the original manuscript to "In addition, considering the amount of available data and topographic conditions, China is divided into nine regions to evaluate the satellite-based HCHO/NO₂ and study the $O_3$ production regimes." in Line 119 of the revised manuscript.

Due to the retrieval uncertainty associated with satellite instruments, the application of satellite-based HCHO/NO₂ is limited to averaging over time or aggregating data with sufficiently large sample sizes to reduce retrieval noise. Based on the dual consideration of the amount of data required to construct the relationship between $O_3$ and satellite HCHO and NO₂ and the amount of work required to process the data, the NO₂ and HCHO columns were sampled every half month and the $O_3$ concentrations at the stations were averaged every half month. We added the relevant information in Line 117 "Considering the data volume and the need to reduce retrieval noise, TROPOMI-retrieved NO₂ and HCHO columns are sampled every half month over $O_3$ sites for the same period."

**12.** *Line 195 – I think it would be more useful to show Figure S2 in place of Figure 3. I am not sure how it is useful to average all the sites together when there are strong regional differences.*

**Response:**

In Figure 3, we first evaluate if satellite-based HCHO/NO₂ can capture the nonlinearities in $O_3$ chemistry, and then derive quantitative relationships between $O_3$ and HCHO/NO₂. After the qualitative approach is established, nine regions are assessed individually and

show regional differences in Table 1 and Figure S2.

**13.** *Line 201 – You say "This varied response is also reflected in different thresholds for other photochemical indicators ... or patterns of $O_3$ isopleths...". Please be specific about what these other studies say and how it fits with your results, right now this sentence is too speculative to be useful.*

**Response: Accepted.**

Our results show that the HCHO/NO$_2$ ranges marking the regime transition vary regionally. Previous studies have also demonstrated regional differences in the threshold values of HCHO/NO$_2$ (Schroeder et al., 2017; Chang et al., 2016) and other photochemical indicators (Liu and Shi, 2021).

We revised the sentences and literature cited in Line 204 to be "Previous studies have also demonstrated regional differences in the threshold values of HCHO/NO$_2$ (Schroeder et al., 2017; Chang et al., 2016) and other photochemical indicators (Liu and Shi, 2021)"

**14.** *Line 202 – You should have the information to actually discuss changes in temperature and radiation across these regions and be a little more specific about whether the could impact HCHO/NO$_2$.*

**Response: Accepted.**

Thanks for your comment. The focus of this study is to derive HCHO/NO$_2$ threshold values marking the regime transition and diagnose ozone-NO$_x$-VOC sensitivity. Our results show that the threshold varies among regions of China. Combined with Liu and Shi's research, this regional difference may reflect environmental conditions. Due to space limitations, we do not discuss this in detail.

**15.** *Line 207 – Give the previously reported values.*

**Response: Accepted.**

The previously reported values were added in Line 210 as follows, "The HCHO/NO$_2$ thresholds in the present study are higher than the previously reported model-based values, such as 1~2 by Jin and Holloway (2015) and Duncan et al. (2010), and 1.5~2.3 by Chang et al. (2016)."

***16.*** *Line 216 – Given that you have shown strong regional differences, I do not think it makes sense to show Fig. S3 at all, and move forward only with regionally-specific analysis.*

**Response:**

Most previous studies derived or used the uniform HCHO/NO$_2$ threshold, but here we show that based on the same HCHO/NO$_2$ thresholds for China and different thresholds for different regions, the two methods yielded somewhat different spatial distributions of O$_3$ sensitivity, thus emphasizing the need to use different thresholds for different regions.

***17.*** *Figure S6 – It is very confusing to have the timeseries split between the supplement and main text. I don't see why putting Fig S6 into Figure 5 and 6 would be problematic.*

**Response:**

The annual maps may not be necessary in order to show general trends. We have replaced Figures 5a and 6a in the revised manuscript to only show 2013, 2017, and 2020. Figures for other years are placed in the supplement.

[Figure]

Figure 5: (a) Maps of average satellite-based NO$_2$ columns over China from April to September, and (b) monthly mean NO$_2$ columns averaged over eastern China and North China Plain in

April-September. Gray shading: mean value ± 50% standard deviation across all grids for each month. Solid line: the linear fitted curve. Inset: absolute annual linear trend and percentage of annual trend (% per year, the linear trend divided by the 2013 mean values).

Figure 6: Same as Figure 5 but for satellite-based HCHO columns.

**18.** *Figure 5b and 6b – Why are you showing annual trends when your ozone analysis is April – September? Stick to analyzing the same time period throughout.*

**Response: Accepted.**

We have replaced Figures 5b and 6b and calculated trends in HCHO and NO$_2$ columns based on April-September each year, as shown in the response to Comment 17.

**19.** *Line 251 – I don't understand the statement "Based on the non-linear relationship of O$_3$–NO$_x$–VOC, the changes in ozone precursors is the direct factor affecting ozone level." Are you trying to argue that other factors like meteorology or changes in PM are not the cause? If so, this statement is not convincing as there has been no discussion of these other factors.*

**Response: Accepted.**

As the first sentence of this section, this is intended to lead to the following discussion of the effect of changes in ozone precursors on ozone concentrations, rather than comparing emissions with other factors. To avoid misunderstandings, this sentence has been revised to "Based on the non-linear $O_3$-$NO_x$-VOC relationship, changes in ozone precursors can directly affect ozone levels." in Line 256 of the revised manuscript.

***20. Figure S7 – Show the same thing for HCHO.***

**Response: Accepted.**

We revised and added trends of satellite $NO_2$ and HCHO columns in Figure S7.

[Figure]

Figure S7. Trends in April-September average surface $NO_2$ concentrations in 2015-2021, and trends in April-September average satellite $NO_2$ and HCHO columns in 2013-2020 with $p < 0.05$.

***21. Figure 6b – Are those trends really statistically significant? Also you need to look at just April-September.***

**Response: Accepted.**

We changed Figure 6b to focus on April-September. The trend of HCHO is statistically insignificant ($p > 0.05$). We have replaced Figure 6b, focusing on April to September each year.

[Figure]

Figure 6: Same as Figure 5 but for satellite-based HCHO columns.

*22. Line 269 – Could it not also be an indicator that as urban ozone titration decreases, and OH increases due to the reduced $NO_x$ level, that oxidation of VOCs increases?*

**Response: Accepted.**

As mentioned by the reviewer, it is possible to increase the yield of HCHO because the oxidation of VOC increases as the $NO_x$ level decreases. In addition, the decrease in $NO_x$ may lead to a decrease in HCHO production from isoprene oxidation (Souri et al., 2020; Wolfe et al., 2016). The available observations are not sufficient to conclusively determine the changes in HCHO production, which requires further investigation. However, the change in HCHO is far from being as sharp as the drop in $NO_2$. We believe this reflects the impact of insufficient VOC reductions in reference to previous studies that used HCHO to indicate changes in VOC (Li et al., 2020; Shen et al., 2019; Zhu et al., 2014).

Considering your comments, we added a sentence in Line 269 "The changes in satellite HCHO can roughly indicate changes in VOC emissions, which has been applied in several previous studies (Li et al., 2020; Shen et al., 2019; Zhu et al., 2014)" We added the relevant discussion in Line 350 of the revised manuscript. "However, the use of satellite

HCHO/NO$_2$ for quantitative diagnosis of ozone-NO$_x$-VOC sensitivity is influenced by the following uncertainties and requires further investigation." "Second, the relationship between satellite-based HCHO and surface VOC reactivity is not fully understood. There are differences in HCHO yields for different classes of VOCs (Shen et al., 2019). Biogenic VOC emissions also have an influence. Since HCHO is a weaker UV-visible absorber than NO$_2$, satellite retrieval of HCHO is more error-prone. These factors may limit their usefulness in detecting HCHO from local sources of anthropogenic VOCs."

**23.** *Line 271 – The authors have not discussed the importance of biogenic isoprene to the VOC budget.*

**Response: Accepted.**

Our results show that there is no significant overall decrease in satellite HCHO concentrations in most of eastern China since 2013, which is similar to the estimated trends of anthropogenic VOC and biogenic VOC emissions in China (Zheng et al., 2018; Simayi et al., 2022; Li et al., 2020). Biogenic VOC emissions are difficult to control, and anthropogenic VOC emissions are much higher than biogenic VOC emissions in urban areas of China, so it is the lack of anthropogenic VOC emission reduction that has not significantly decreased HCHO.

We revised the sentence in Line 277 to be "In addition to anthropogenic and biogenic emissions, the long-term changes in HCHO are driven by several other factors ...". We added a sentence in Line 294 "Considering that biological VOC also increased, it is clear that the reduction of anthropogenic VOCs is not sufficient to bring down the total VOC emissions."

**24.** *Line 273 – Show us a temperature timeseries.*

**Response: Accepted.**

We supplemented the time series of monthly mean HCHO and temperature in the Supplement as Figure S9.

[Figure]

Figure S9. Time series of monthly mean HCHO columns and temperature in eastern China from April-September 2013-2020.

(Temperature data source: Yearbook of Meteorological Disasters in China)

*25. Line 311 – The authors are assuming that HCHO columns directly relate to VOC emissions, while changes in chemistry due to NOₓ reductions could have a large impact (see my above comment). Thus I do not think you can state that the increase in HCHO is due to increased VOC emissions without further evidence.*

*The authors need to explain how the "vigorous development of stall businesses" could cause this increase.*

**Response: Accepted.**

In this case, there is a slight increase in surface $NO_2$ and a slight decrease in satellite $NO_2$, but a large increase in HCHO in Chengdu. As mentioned by the reviewer, it is possible to increase the yield of HCHO because the oxidation of VOC increases as the $NO_x$ level decreases. In addition, the decrease in $NO_x$ may lead to a decrease in HCHO production from isoprene oxidation (Souri et al., 2020; Wolfe et al., 2016). The available observations are not sufficient to conclusively determine the changes in HCHO production. However, in general, the variation of HCHO is much larger than that of $NO_2$, which is contributed by the increase of VOC. We added a sentence in Line 269 "The changes in satellite HCHO can roughly indicate changes in VOC emissions, which has been applied in several previous studies (Li et al., 2020; Shen et al., 2019; Zhu et al., 2014)." We added the

relevant discussion in Line 350 of the revised manuscript. "However, the use of satellite HCHO/NO$_2$ for quantitative diagnosis of ozone-NO$_x$-VOC sensitivity is influenced by the following uncertainties and requires further investigation." "Second, the relationship between satellite-based HCHO and surface VOC reactivity is not fully understood. There are differences in HCHO yields for different classes of VOCs (Shen et al., 2019). Biogenic VOC emissions also have an influence. Since HCHO is a weaker UV-visible absorber than NO$_2$, satellite retrieval of HCHO is more error-prone. These factors may limit their usefulness in detecting HCHO from local sources of anthropogenic VOCs."

The vigorous development of stall businesses led to VOC emissions from open air restaurant sources, although specific emissions are difficult to obtain. We revised this sentence in Line 320 "At that time, after the strictest restrictions, Chengdu was vigorously developing the stall business to help work resumption. The increase in VOCs may be related to open-air cooking emissions (Liang et al., 2022)."

**26.** *Line 325 – The authors haven't discussed PM$_{2.5}$ at all to support this statement.*

**Response: Accepted.**

In the section of the Introduction, we mention the previously studied contribution of PM$_{2.5}$ decline to O$_3$ pollution and its controversy. The following relevant discussion has been added to Line 368 of the revised manuscript, and we have revised the title of Section 4 to "Discussion".

"Some studies suggested that the significant decrease in PM$_{2.5}$ is the most crucial factor contributing to the O$_3$ increment in China (Li et al., 2019), but the idea of the impact of HO$_2$ uptake is controversial (Tan et al., 2020). The summertime MDA8 O$_3$ enhancement due to changes in PM$_{2.5}$ levels in NCP during 2013-2017 estimated by Li et al. (1 ppb year$^{-1}$) is insufficient to explain the observed trend (3.3 ppb year$^{-1}$) (Lu et al., 2020), which indicates that the effect of PM$_{2.5}$ is not the essential cause of worsening ozone pollution. The simultaneous decline in O$_3$ and PM$_{2.5}$ concentrations in Beijing in recent years (Ren et al., 2021) suggests that appropriate VOC and NO$_x$ emission reductions can control both PM$_{2.5}$ and O$_3$ pollution."

**27.** *Figure 8 – I don't understand the purpose of this figure.*

**Response: Accepted.**

Due to the similarity of Fig. 3a with the O$_3$-VOC-NO$_x$ isopleths, we try to infer the ridges of the O$_3$-HCHO-NO$_2$ relationship for cities to obtain the ratio of HCHO/NO$_2$ reduction

that causes the fastest decrease in $O_3$ concentration.

[revised manuscript text omitted]

---

## Author Response (AR2)

**Author's Response**

Dear Editor,

We are delighted to have the opportunity to resubmit a revised manuscript and would like to thank you very much for your help throughout the process. We would also like to sincerely thank the reviewer for the valuable suggestions, which we believe helped to significantly improve the quality of the manuscript. Detailed responses to the reviewer's comments are listed below, and the corresponding changes have been made in the revised manuscript based on these comments. All changes are marked in blue.

**Response to Referee #1:**

*The authors made substantial revisions that address many of my concerns in the first round of review. I will outline several remaining comments below for the authors and the editor to consider, and I support the publication of the paper if they can be addressed.*

**Response:**

We thank the reviewer for the comments and suggestions, which have been very helpful in improving the manuscript. The responses and revisions are listed below.

*1) Use of two satellite instruments. First, more details should be included to describe the "merging" process, either in the main text or in the supplement. To what extent did the OMI and TROPOMI data differ during 2018-2020, and how this differences were minimized during the "merging"?*

*Secondly, does the NASA official OMI retrievals contain data up to 2021 to verify the trends in this paper from an "independent" perspective? The retrieval from the NASA official products are different although the same OMI radiance is used, but the consistent OMI instrument throughout the period will provide valuable verification, at least for the relative/qualitative trends. Such rigor is needed for trends from two instruments.*

*Finally, the TROPOMI resolution (0.01 deg) as described by the authors looks unusual, and should be checked to confirm.*

**Response: Accepted.**

To avoid differences between the two satellite instruments, we replaced the official NASA OMI retrieval data covering 2013-2021 in the revised manuscript to study the long-term changes in $NO_2$ and HCHO. Tropospheric $NO_2$ vertical columns were obtained from OMI/Aura Level-3 $NO_2$ products (OMNO2d 003) with a grid resolution of 0.25°×0.25°, while HCHO total columns were obtained from OMI/Aura Level-3 HCHO products (OMHCHOd 003) with a grid resolution of 0.1°×0.1°. We recalculated the trends. Although the values have changed compared to the previous manuscript, the conclusions remain consistent.

The high spatial resolution TROPOMI data is still used to derive the HCHO/$NO_2$ threshold values marking transitions in $O_3$ formation regimes and to diagnose the ozone-$NO_x$-VOC sensitivity in China. After verification, the TROPOMI NO2 and HCHO resolution from Earth Engine is 1113.2 meters, which is about 0.009° in the China region. We have made the corresponding changes.

We revised the relevant description in Lines 101-109:

"The TROPOMI data from the Earth Engine Data Catalog are based on the algorithm developments for the QA4ECV reprocessed dataset for OMI and have been further optimized. The TROPOMI data, available for 2019-2021, are processed with a spatial resolution of 1113.2 meters (about 0.009∘ within China). The same chemistry transport model for HCHO and $NO_2$ is better suited for analyzing their ratio than products developed with different prior profiles.

The OMI data with longer time horizons (2013-2021) are used to study the long-term changes in $NO_2$ and HCHO and track changes in emissions of $NO_x$ and VOCs. Tropospheric $NO_2$ vertical columns are obtained from OMI/Aura Level-3 $NO_2$ products (OMNO2d 003) with a grid resolution of 0.25°×0.25°, while HCHO total columns were obtained from OMI/Aura Level-3 HCHO products (OMHCHOd 003) with a grid resolution of 0.1°×0.1°. Since HCHO mainly presides in the troposphere, its total column can be regarded as the tropospheric column (Duncan et al., 2010)."

We revised Figures 5 and 6, as well as the data results in the manuscript.

[Figure]

**Figure 5: (a) Maps of average satellite-based $NO_2$ columns over China from April to September, and (b) monthly mean $NO_2$ columns averaged over eastern China and North China Plain in April-September. Gray shading: mean value ± 50% standard deviation across all grids for each month. Inset: absolute annual linear trend and percentage of annual trend (% per year, the linear trend divided by the 2013 mean values).**

[Figure]

**Figure 6: Same as Figure 5 but for satellite-based HCHO columns.**

*2) Since this paper extends previous investigations, it will be valuable to include one paragraph in the discussion section to summarize comparison of the findings in this paper vs. the previous ones, and highlight new findings and implications from this study.*

**Response: Accepted.**

To emphasize the new findings and significance of this study, we made additions and changes in Lines 313-324 of the revised manuscript,

"Based on the evaluation of the nonlinearity of $O_3$-$NO_x$-VOC chemistry captured by space-based $HCHO/NO_2$, this study derives thresholds marking the transitions between chemical regimes in different regions of China and diagnoses the current spatial distribution of $O_3$ production regimes. To reveal the causes of $O_3$ increases, $O_3$ responses to precursors changes are evaluated by tracking VOCs and $NO_x$ with satellite HCHO and $NO_2$.

Results showed that the $HCHO/NO_2$ ranges of transition from VOC-limited to $NO_x$-limited regimes vary apparently among Chinese regions, which is inconsistent with previous studies (Wang et al., 2021; Li et al., 2021). The higher resolution TROPOMI satellite data enables a better match between the location of ground-based $O_3$ monitoring stations and the grid of satellite data, thus allowing a more accurate derivation of $HCHO/NO_2$ thresholds and reflection of ozone-$NO_x$-VOC sensitivity. For April-September 2021, VOC-limited regimes

are widely found over megacity clusters (NCP, YRD, and PRD) and concentrated in developed cities (such as Chengdu, Chongqing, Xi'an, and Wuhan). $NO_x$-limited regimes dominate most of the remaining areas. Moreover, the high-resolution TROPOMI data can more accurately resolve strongly VOC-limited conditions in urban cores."

and in Lines 333-348:

"Identifying the causes of ozone increases since 2013 is crucial for effectively controlling the ozone pollution in China. From 2013 to 2021, satellite $NO_2$ and HCHO columns showed an annual decrease of 3.0% and 0.3%, respectively, indicating an effective reduction in $NO_x$ emissions alone without effective VOC control. …

In summary, the significant reduction in $NO_x$ alone without effective control of VOC, combined with the effect of the decrease in $PM_{2.5}$ mentioned in previous studies (Li et al., 2019; Li et al., 2022), has led to an increase in $O_3$ in major regions in China."

**3)** *Section 3.4 still reads loosely connected with the previous results to me. The selection of locations and time (e.g. April in Beijing and May in Chengdu) reads random, and unrepresentative of COVID. My suggestion is this section can be cut. The authors can develop this idea into an independent paper. I leave this comment to the editor to consider.*

**Response: Accepted.**

Considering the consistency of the topic related to long-term changes, we have removed the section on cases during COVID-19 in the revised manuscript.

**4)** *Line 368-373: In the Li et al. (10.5194/acp-20-11423-2020, 2020) paper, the ozone trends over NCP (2013-2019) is 3.3 ppb/yr, in which 1.4 ppb/yr is attributed to meteorology. So I do not support the authors description here by simply comparing numbers. 1 ppb/yr is a significant component of 3.3 ppb/yr. I suggest to revise these words to state that "PM, VOC, and NOx are all very important anthropogenic drivers of ozone trends in China", which is also supported by a recent paper (Li et al., 10.1021/acs.est.2c03315, 2022).*

**Response: Accepted.**

Our study highlights that the increase in ozone in major areas is due to significant reductions in $NO_x$ only without effective VOC control. In the revised manuscript, we modified the relevant expressions as follows in Lines 346-348: "In summary, the significant reduction in $NO_x$ alone without effective control of VOC, combined with the effect of the decrease in

PM$_{2.5}$ mentioned in previous studies (Li et al., 2019; Li et al., 2022), has led to an increase in O$_3$ in major regions in China."

We also revised the relevant expressions in the Introduction (Lines 59-61): "Recent O$_3$ trends in China have been driven by a variety of anthropogenic factors. Some model simulations revealed a strong influence of the PM$_{2.5}$ decrease on the O$_3$ increase and attributed that response to the aerosol sink of hydroperoxy (HO$_2$) radicals (Li et al., 2019)."